



# Introducing a VIIRS-based Fire Emission Inventory version 0 (VFEIv0)

Gonzalo A. Ferrada[1], Meng Zhou[2], Jun Wang[1,2], Alexei Lyapustin[3], Yujie Wang[4], Saulo R. Freitas[5], Gregory R. Carmichael[1]

[1]Center for Global and Regional Environmental Research, The University of Iowa, Iowa City, IA, USA
[2]Interdisciplinary Graduate Program in Informatics, The University of Iowa, Iowa City, IA, USA
[3]Laboratory for Atmospheres, NASA Goddard Space Flight Center, Greenbelt, MD, USA
[4]Joint Center for Earth Systems Technology, University of Maryland Baltimore County, Baltimore, MD, USA
[5]Center for Weather Forecast and Climatic Studies, National Institute for Space Research, São José dos Campos, SP, Brazil

*Correspondence to*: Gonzalo A. Ferrada (gonzalo-ferrada@uiowa.edu)

**Abstract.** A new open biomass burning inventory is presented that relies on the fire radiative power data from the Visible Infrared Imaging Radiometer Suite (VIIRS) onboard the Suomi NPP satellite. This VIIRS-based Fire Emission Inventory (VFEI) provides emission data from early 2012 to 2019 for more than 40 species of gases and aerosols at spatial resolutions of around 500 m. We found that VFEI produces similar results when compared to other major inventories in many regions of the world. Additionally, we conducted regional simulations using VFEI with the Weather Research and Forecasting (WRF) model with chemistry (WRF-Chem) for Southern Africa (September 2016) and North America (July-August 2019). We compared aerosol optical depth (AOD) from the model against two observational datasets: the MODIS Multi-Angle Implementation of Atmospheric Correction (MAIAC) product and AErosol RObotic NETwork (AERONET) stations. Results showed good agreement between both simulations and the datasets, with mean AOD biases of around +0.03 for Southern Africa and –0.01 for North America. Both simulations were not only able to reproduce accurately the AOD magnitudes, but also the inter-diurnal variations of smoke concentration. In addition, we made use of the airborne data from the ObseRvations of Aerosols above CLouds and their intEractionS (ORACLES; Southern Africa) and the Fire Influence on Regional to Global Environments Experiment and Air Quality (FIREX-AQ; North America) campaigns to evaluate the simulations. In Southern Africa, results showed correlations higher than 0.77 when comparing carbon monoxide and black carbon. In North America, correlations were lower and biases higher. However, this is because the model was not able to reproduce the timing, shape, and location of individual plumes over complex terrain (Rocky Mountains) during the FIREX-AQ campaign period.

## 1 Introduction

Biomass burning (BB) occurs in all vegetated regions of the world. Fires can be ignited naturally through lightning or via anthropogenic sources. For example, in tropical regions, fires are used crop clearing or deforestation. Based on different datasets, Andreae (2019) estimated that between 4.3 and 12.9 Pg a$^{-1}$ of dry matter is burned globally, which leads to the emission of several types of aerosol species and many trace gases. Many of these emissions, such as carbon dioxide ($CO_2$),



water vapor, methane ($CH_4$), and black carbon (BC), among others, play an important role in global warming. Bond et al. (2013) estimated that the radiative forcing of BC is +1.1 W m$^{-2}$ and that BB is the source of between 33 and 41 % of the global BC emissions. Fires also emit several ozone precursors (hydrocarbons and nitrogen oxides) that combined with the emitted
particulate matter create poor air-quality episodes and health emergencies when sensitive populations are exposed (e.g., Dennekamp et al., 2015; Knorr et al., 2017). Moreover, aerosol particles can act as cloud condensation nuclei, altering the physics of clouds and precipitation patterns. Thus, aerosols have not only a direct effect on radiation, but indirect and semi-direct effects as well (Ge et al., 2014).

Several BB inventories have been created to quantify emissions of diverse species in different biomes. They are useful for
atmospheric composition, weather, and climate studies, often using atmospheric models. These inventories rely on remote sensing observations of thermal anomalies or "hotspots". Some of them, e.g., the Global Fire Emissions Database (GFED; van der Werf et al., 2017, 2010), the Fire INventory from NCAR (FINN; Wiedinmyer et al., 2011) and the Fire Locating and Modeling of Burning Emissions (FLAMBE; Reid et al., 2009) use the *bottom-up* approach which consists of using the burned area to calculate emissions for species $s$ ($E_s$) as follows:

$$E_s = M_{dry}\,\beta\,A_{burnt}\,EF_s, \tag{1}$$

where $M_{dry}$ represents the biomass above ground available to burn, $\beta$ is the combustion factor, i.e., the fraction of the biomass that burns, $A_{burnt}$ is the burnt area and $EF_s$ is the emission factor for a given species. The burnt area is estimated from satellite data, while the other coefficients come from literature and often are categorized on the biome where the fire occurs. However, some authors (e.g., Hoelzemann et al., 2004; Ito and Penner, 2004; Ichoku and Kaufman, 2005) have noted that the burnt area
estimations can have large uncertainties because it is a variable difficult to measure.

Other inventories make use of the fire radiative power (FRP) which can be defined as the part of chemical energy emitted as radiation during the BB process and it is indicative of the fire intensity. In this case, emissions for each species are calculated as follows:

$$E_s = \kappa\,EF_s\,\frac{FRP}{A}, \tag{2}$$

where $A$ is the satellite pixel area and $\kappa$ is an empirical constant that relates the FRP to the amount of dry biomass burnt. The ratio between $FRP$ and $A$ is often called FRP density. This technique is called *top-down* approach and examples of emission inventories in this category are the Quick Fire Emission Dataset (QFED; Darmenov and da Silva, 2013), the Global Fire Assimilation System (GFAS; Kaiser et al., 2012) and the Fire Energetics and Emissions Research (FEER; Ichoku and Ellison, 2014). While almost all inventories rely on observations from the Moderate Resolution Imaging Spectroradiometer (MODIS)
onboard of the Terra and Aqua satellites, they have large uncertainties, sometimes up to a factor of 10 at the regional scale (Zhang et al., 2014; Wang et al., 2018; Pan et al., 2019).

With the increase of computational resources available, atmospheric models now include sophisticated chemistry and aerosol mechanisms, considering hundreds of species and reactions, and are capable of using high-resolution model grids for local studies (e.g., Grell et al., 2005; Emmons et al., 2010; Pfister et al., 2020; Bindle et al., 2021; Bey et al., 2001; Freitas et al.,





2017). However, BB emission inventories often provide data for a limited number of species and at spatial resolutions no higher than 0.1° (~10 km). These can be a problem in model simulations since missing species can be highly volatile (e.g., hydrocarbons) and, thus, producing unrealistic results in smoke plume composition. In addition, coarse resolution inventories can produce a misplacement of fires in high-resolution model grids (< 10 km) since the model matches the closest coordinates from the inventory into the grid (see section 5). This can make the smoke plumes and its transport to be subject to different

circulation patterns which is particularly important in regions of complex topography.

   Aiming to solve these potential issues, in this paper we introduce an open biomass burning emission inventory using the *top-down* approach. We use FRP estimations from the Visible Infrared Imaging Radiometer Suite (VIIRS) instrument I-band (Schroeder et al., 2014) onboard of the Suomi National Polar-orbiting Partnership (Suomi NPP) and NOAA-20 satellites. We named this inventory as VIIRS-based Fire Emission Inventory version 0 (VFEIv0; hereafter simply as VFEI). VFEI provides

daily emission fluxes for 46 species of gases and aerosols using emission factors from Andreae (2019) at a resolution of 0.005° (~500 m). The paper is organized as follows: section 2 describes the methodology of VFEI, including assumptions and filters applied. In section 3 we present VFEI product results and comparison to other major BB inventories. Section 4 describes the atmospheric simulations made with the Weather Research and Forecasting (WRF) model with chemistry (WRF-Chem; Grell et al., 2005) for two regions: Southern Africa in September 2016 and North America in July-August 2019. Model results are

compared against aerosol optical depth (AOD) observations from the Moderate Resolution Imaging Spectroradiometer (MODIS) Multi-Angle Implementation of Atmospheric Correction (MAIAC; Lyapustin et al., 2018) product and AErosol RObotic NETwork (AERONET). In addition, a comparison is made against airborne data of aerosol and gaseous species from the ObseRvations of Aerosols above CLouds and their intEractionS (ORACLES; Redemann et al., 2020) and the Fire Influence on Regional to Global Environments Experiment and Air Quality (FIREX-AQ; https://www.esrl.noaa.gov/csd/projects/firex-

aq/, last access: 21 April 2022) field experiments carried out in the South Atlantic Ocean and the Continental US, respectively.

## 2 Description of VFEI

   VFEI produces emissions by applying the *top-down* approach as shown in Eq. (2), where $EF_s$ has units of g [species] kg⁻¹ [dry-matter], FRP has units of MW (or MJ s⁻¹), $A$ is in m² and $\kappa$ in kg [dry-matter] MJ⁻¹. The FRP information is taken from the VIIRS active fire detection data product (Schroeder et al., 2014; Csiszar et al., 2014) described in section 2.1. Table 1 shows

the 46 species of gases and aerosols that VFEI considers and their emission factors ($EF_s$). These emission factors were compiled by Andreae (2019) using data available from many studies related to open biomass burning emissions, including field campaigns, and considers six different biomes. These emission factors are an update of an older version presented by Andreae and Merlet (2001). The emission factor of nitrous oxide (N₂O) over peatland was filled using its value from savanna and grassland, since it was not reported in the source. In addition, the conversion factors ($\kappa$) used in VFEI vary according to

the biome (see table 2 of Kaiser et al., 2012). These conversion factors were developed by Heil et al. (2010). They made a linear regression analysis between the dry matter combustion from GFEDv3.1 and MODIS FRP to produce these coefficients,



which are also applied in the GFAS inventory. Both emission and conversion factors are assumed to be climatologically representative for each biome type.

## 2.1 VIIRS active fire product

VFEI uses the daily hotspot information taken from the VIIRS active fire detection data product (Schroeder et al., 2014; Csiszar et al., 2014; doi:10.5067/VIIRS/VNP14IMG.001). The algorithm uses a similar approach as in the MODIS Fire and Thermal anomalies product (Giglio et al., 2003) but relies on the middle and thermal infrared imagery data from VIIRS I-band which has a resolution of 375 m at nadir. This high-resolution allows for a better detection of small fires when compared against VIIRS M-band (750 m nominal resolution; Schroeder et al., 2014) and MODIS (1 km resolution at nadir; Kaufman et al.,

1998). Another advantage of VIIRS over MODIS is the wider swath of observation. The swath width of MODIS is around 2340 km while VIIRS has a swath width of ~3040 km. This means that VIIRS offers daily observations with no data gaps over the tropics. In addition, Li et al. (2018) made a comparison between the active fire products from MODIS and VIIRS and found that the FRP values are comparable between them with small variations in some regions. They further noted that MODIS is strongly dependent on the view zenith angle (i.e., pixel growth with scan angle), while VIIRS, due to its different design,

show much less dependance (Wang et al., 2018). Furthermore, VIIRS mid-infrared I-band has a lower saturation temperature than MODIS and VIIRS M-band (Polivka et al., 2015), and thus, the FRP estimations for the VIIRS I-band product is derived from VIIRS M-bands and equally split into the observed I-band pixels.

Currently, VIIRS is onboard two polar orbiting satellites: (1) Suomi-NPP, a joint mission between the National Aeronautics and Space Administration (NASA) and the National Oceanic and Atmospheric Administration (NOAA), was launched in

October 2011 and began providing data since early 2012; and (2) NOAA-20 which was launched in November 2017; however, data began to be available since early 2020. Here, we focus on the data from VIIRS onboard Suomi-NPP because we make analysis and comparison to other inventories during a period of 8 years (from 2012 to 2019) and the VIIRS active fire detection data product was developed based on the instrument onboard Suomi-NPP. In addition, the fire detection data product from NOAA-20 was not available as of mid-202. Suomi-NPP overpasses the equator at around 1:30 PM and AM local time.

## 2.2 Gridding of hotspots

From the VIIRS active fire product, VFEI uses the information on coordinates (latitude and longitude), FRP, pixel area ($A$), the day/night flag and the confidence (low, nominal, or high) of each hotspot. The confidence is an indication of the quality of individual hotspots or fire pixels, and it is determined within the active fire algorithm. Low confidence pixels are associated with areas contaminated by sun glint and low relative temperature anomaly (<15 K) in the middle infrared channel (I4).

Nominal and high confidence observations are free of sun glint contamination during daytime and, in contrast, have a high temperature anomaly in the mid-infrared channel. Furthermore, high confidence pixels are characterized by day- or nighttime saturated pixels. To assure the quality of the emissions and remove potential false detections, VFEI does not consider low confidence observations.



The day/night flag is used to separate the hotpots into daytime and nighttime observations. Then, each hotspot is gridded into
a global fine grid of 0.005° spatial resolution in both latitude and longitude directions. This is done by using the nearest
neighbor remapping approach and the purpose is to reduce the potential multiple observations of the same fire that can occur
around the edges of the swath and/or overlapping areas of different swaths (i.e., bowtie effect; Polivka et al., 2016). If a single
grid point of VFEI contains more than one observation, then the average values of FRP and pixel area are taken. After gridding
the hotspots separately for day- and nighttime observations, we take the average between both. This is done to constrain the
average daily emission fluxes that are reported by VFEI. The assumption behind this is that fires tend to have a peak on FRP
around early afternoon and a minimum during the night, following a gaussian distribution (Darmenov and da Silva, 2013;
Kaiser et al., 2012; Pereira et al., 2009; Roberts et al., 2009; Li et al., 2019). The result is a single fine grid with average daily
FRP density (FRP/pixel area) that include a representation of the typical diurnal cycle of fires.

### 2.3 Biome map

The emission factors from Andreae (2019; table 1) are classified into 6 biomes: tropical forest, temperate forest, boreal forest,
savanna and grassland, cropland and peatland. A common way to categorize each fire into these biomes is to use MODIS Terra
and Aqua combined land cover product (Friedl et al., 2015; doi:10.5067/MODIS/MCD12C1.006), which contains 17 land
categorizations defined by the International Geosphere-Biosphere Programme (IGBP). This technique has been used in FINN,
QFED and GFAS, among others. However, there is not a consensus on how to differentiate between tropical, temperate, and
boreal biomes. For example, QFED defines forests between 30° S and 30° N as tropical forests, and FINN defines as boreal to
all forests at latitudes higher than 50° N. On the other hand, Andreae (2019) suggests that boreal forests exist at latitudes higher
than 60° N. These rough estimations on the transitions of the biomes could introduce uncertainties on the final emissions
reported. To address this, we not only use the land cover categorizations from MODIS, but also complement it with the Köppen
climate classifications (Beck et al., 2018) to differentiate between tropical, temperate and boreal biomes.

Table 2 shows the mapping of the 17 land cover categories from MODIS and the Köppen climate classifications used to define
each biome considered in VFEI. We selected 6 warm climates to define a tropical layer and 8 cold climates to define a boreal
layer (Fig. S1). The boreal mask was limited to the northern hemisphere only. Then, the biomes in VFEI were defined when
both conditions (IGBP land cover and climate) were met. The global result is shown in figure 1 and this biome map provides
more realistic transitions among tropical, temperate, and boreal regions.

### 2.4 Static corrections

Some filters are applied to the data from the VIIRS active fire product to reduce uncertainties and false detections. Besides the
bowtie correction to avoid potential overrepresentation of fires, we also remove data from gas flares using the dataset presented
by Elvidge et al. (2016), which contains coordinates of every gas flare in the world (Fig. S1). Thus, hotspots within a radius
of ~2 km of these gas flares are removed from the processing before gridding.



In addition, after gridding, all hotspots over water are inspected individually. We examine the 8 adjacent grid points of each fire over water to determine if there is land around it. If at least one of these adjacent grid points is land, then the pixel in question is kept. The assumption here is that the fire occurred near a body of water and this situation can happen because VFEI uses the dominant IGBP land categories from MODIS which can hide potential fractions of land in such grid point. On the other hand, if none of the adjacent pixels is land, then the fire in question is removed from the dataset.

Another important correction is related to the VIIRS FRP data. As mentioned in section 2.1, VFEI only uses data from VIIRS onboard of Suomi-NPP, which overpasses the equator at around 1:30 AM/PM. In contrast, MODIS is available onboard of Terra and Aqua satellites that overpass the equator at around 10:30 and 1:30 AM/PM, respectively. Then, MODIS provides more data to constrain the diurnal cycle of the fires and, thus, the total daily emissions. Because of this, we scale up the FRP from VIIRS by a factor of 1.2 globally. This factor was determined by using the same methodology described here but using

data from the MODIS active fire product (Giglio et al., 2003, 2016). The results (not shown) indicated that global emissions with MODIS were 20 % higher than the ones with VIIRS. This factor is also in agreement with Li et al. (2018).

### 2.5 Product

The VFEI product is offered in NetCDF format (version 4) that includes daily global emission fluxes (in kg m$^{-2}$ s$^{-1}$) for the species defined in table 1 at the nominal resolution of 0.005° (~500 m). Each hotspot contains information on grid geolocation

(latitude and longitude), plus average FRP (MW). Other supplemental information, such as area of the VFEI grid cell and molecular weight of gaseous species are included in case a unit conversion is needed by the user. The file size of each daily NetCDF is around 5 to 15 Megabytes, depending on the number of hotspots included in a particular day. The data ranges from 20 January 2012 to present, except where the VIIRS active fire product was not available, i.e., 25 March 2012 and 25 August 2014. The most recent data is classified as near-real-time (NRT), while the data older than 3 months is classified as archived,

which contains all the quality assurance provided in the VIIRS active fire product. The data can be downloaded from a dedicated server from the University of Iowa (http://bio.cgrer.uiowa.edu/VFEI/DOWNLOAD; last access: 21 April 2022).

### 3 Results

### 3.1 Global emissions

According to VFEI, open BB emits 2098.5 Tg of carbon globally every year. We expressed carbon emissions by taking the

carbon part of five dominant species: carbon monoxide (CO), $CO_2$, $CH_4$, black carbon and organic carbon (OC):

$$E_C = \frac{12}{28}E_{CO} + \frac{12}{44}E_{CO_2} + \frac{12}{16}E_{CH_4} + E_{BC} + E_{OC} \,, \tag{3}$$

where the ratios represent the weights of carbon on each species (molar mass of carbon over molar mass of compound). However, emissions vary greatly from region to region (Fig. 2a). Table 3 shows the 10 regions defined for analysis that are also drawn in figure 2a. Thus, tropical regions (CAm, SAm, NAf, SAf and SEA) account for more than 78 % of carbon



emissions, where Southern Africa alone accounts for 35.9 % (754 Tg a$^{-1}$) of the global total. Furthermore, each of these regions show a well-defined annual cycle (Fig. 2b), where tropical regions tend to have a peak of BB during their respective dry seasons, while in extratropical regions the peak is during spring and summer. Figure 2b also shows the interannual variability of the fire activity in every region. Both African regions (NAf and SAf) and Central America (CAm) have a low year-to-year variability, because most of the fires in these regions are man-made agricultural fires. On the other hand, extratropical regions,

such as Siberia (Sib), North America (NAm) and boreal America (BAm), show a high interannual variability. This can be attributed to the nature of the fires in these regions, in which natural ignition of fires via lighting is more common. Thus, being subject to interannual climate variabilities.

**3.2 Comparison to other inventories**

We compared VFEI budgets against other four major inventories detailed in table 4. The versions that we used for comparison

are GFED4.1s, FINNv1.5, QFEDv2.5r and GFASv1.2. These inventories rely in one or more compilations of emission factors for their species and biomes considered. All these inventories use hotspots detections from MODIS and key differences among their methodologies are that GFED and FINN use the burnt area to estimate emissions (*bottom-up* approach), while QFED and GFAS use the FRP (*top-down* approach). These inventories were chosen because they are widely used, especially in the models of their respective institutions developed them, and they are easily accessible on the internet. For simplicity, hereafter, we refer

to them as their core name only, omitting their versions, except for FINN because we additionally include a recently developed version from this inventory (FINNv2.4). FINNv2.4 also uses the bottom-up approach but combines both MODIS and VIIRS hotspots detections to generate emissions. Furthermore, this newest version includes improvements in the fuel loadings and emission factors; however, a new version to be released in the future will include more developments and adjustments to specific regions (Christine Wiedinmyer, personal communication).

Figure 3 shows the average annual emissions (2012-2019) of selected species from VFEI, and each inventory mentioned above for each region displayed in figure 2. To make inventories comparable among each other, we applied the definition of total carbon from Eq. (3) to all inventories. Globally, all of them, including VFEI, seem to agree that the total carbon emissions are around 2000 Tg a$^{-1}$, except for FINNv2.4 which doubles that amount (Fig. 3a). The same pattern is observed in the other selected species (CO, SO$_2$, BC, OC and PM$_{2.5}$), however, QFED enhances emissions of the last four species by a factor of up

to 5 depending on the biome (Darmenov and da Silva, 2013). Because of this, QFED emissions of these four species are always higher than the other inventories in all regions.

When BB emissions are analyzed by region, inventories tend to disagree more among them. This happens especially in regions with more uncertain emission factors, such as boreal forests (Sib and BAm) and peat fires that occur usually in regions of Southeast Asia (SEA). Because of this, inventories often apply adjustment factors to cope with aerosol loadings observed in

these regions, producing differences among them that can vary up to factor of 7 in total carbon and up to 10 in BC at the regional level. In general, VFEI emissions are lower than the average of the other inventories in such regions of high uncertainty, and this could be because VFEI does not include a treatment for fires obscured by clouds, which is particularly





important at high latitudes. Thus, it is not a surprise that VFEI has lower emissions than the other inventories in Sib and BAm. In addition, VFEI does not apply any adjustment factor depending on biome, since more global modeling studies are needed.

As implied in section 3.1, the major global sources of BB carbon emissions are SAf (35.9 %), NAf (18.7 %) and SAm (10.9 %). In these regions, inventories tend to agree among each other (except FINNv2.4 and QFED because of the reasons explained above). Nonetheless, VFEI estimates higher emissions in SAf than the other inventories with 754 Tg C a$^{-1}$, which is around 16 % higher than the carbon emissions estimated by GFED (649 Tg C a$^{-1}$) and nearly 50 % more than the estimations by QFED and GFAS (550 and 498 Tg C a$^{-1}$, respectively). The higher emissions in VFEI may be because most of the fires in SAf tend

to be small fires and VIIRS has a higher capability than MODIS on detecting them due to its higher resolution. In section 4, we conducted atmospheric model simulations for Southern Africa to examine the accuracy of VFEI.

It is important to note that all these inventories, including VFEI, are estimations and none of them are observations of actual BB emissions. Each of the inventories have their own uncertainties and assumptions that explain most of the differences among them and most of the emission factors used are generalizations from lab experiments and in-situ measurements. However, as

explained by several authors (e.g., Andreae and Merlet, 2001; Akagi et al., 2011; Andreae, 2019) these observations used to compile the emission factors are scarce in space and time.

## 4 Atmospheric model simulations

### 4.1 Model setup

To assess VFEI emissions, we conducted atmospheric simulations using WRF-Chem (Grell et al., 2005) version 4.2 for two

domains (Fig. 4): (1) Southern Africa during September 2016 at 28 km spatial resolution and (2) North America between 20 July and 11 August 2019 at 20 km resolution. The North American simulation also considered a nested domain over the Pacific Northwest at 6.7 km resolution. All simulations considered 52 vertical layers, with higher resolution near surface. For all domains, outputs were produced every three hours containing information on meteorology, chemistry and aerosols and every one hour for optical properties, such as AOD. Hereafter, we refer to these simulations as WRF-VFEI. It is important to note

that these simulations allow for an assessment on the quantities of VFEI emissions in typical model mid-resolution applications. However, they do not exercise the advantages of the high-resolution of VFEI. Section 5 provides an insight of this capability.

The domains and periods of these simulations were chosen to overlap with two recent field experiments: ORACLES (Redemann et al., 2020) and FIREX-AQ (https://www.esrl.noaa.gov/csd/projects/firex-aq/, last access: 21 April 2022). Both

experiments were focused on measuring smoke composition using aircraft equipped with diverse instruments. ORACLES used the P-3 plane from the National Aeronautics and Space Administration (NASA); however, all their flights were conducted over the South Atlantic Ocean (Fig. 4a) because its main focus was to study the interaction of smoke with the semi-permanent stratocumulus layer over the ocean. During 2016, ORACLES was based in the airport of Walvis Bay, Namibia. Similarly, the NASA DC-8 aircraft was used during FIREX-AQ to sample fresh and semi-fresh smoke plumes from wildfires and agricultural



fires in the Continental US. During its first phase (22 July to 17 August 2019), FIREX-AQ was based in Boise, Idaho, USA, with most of the flights conducted over the Pacific Northwest region (Fig. 4b). In its second phase, FIREX-AQ was deployed in Salina, Kansas, until early September 2019; in this study we focused only in its first phase.

Both simulations were made using meteorological data from ERA5 (Hersbach et al., 2020) and chemistry and aerosol data from the Copernicus Atmosphere Monitoring Service (CAMS) reanalysis (Inness et al., 2019) as initial and boundary

conditions. Both datasets are produced by the European Centre for Medium-Range Weather Forecasts (ECMWF). We allowed a 14-day period for *spin-up* of the model from its initial condition. As mentioned above, simulations for North America were made using a nested domain with continuous feedback and online with its parent domain.

The chemistry mechanism used in both simulations was the Model for Ozone and Related chemical Tracers T1 (MOZART-T1; (Emmons et al., 2020) complemented with the Model for Simulating Aerosol Interactions and Chemistry (MOSAIC; Fast

et al., 2006; Zaveri et al., 2008; Gao et al., 2016). MOSAIC uses a sectional representation of aerosol size distribution, and considers interactions with radiation and clouds (Chapman et al., 2009). MOSAIC provides aerosol data in eight size bins. The chemical constituents of aerosols are assumed internally mixed within each bin; however, they are considered externally mixed between bins. Other selected model configurations are shown in table S1.

**4.2 Emissions**

We used biogenic emissions by using Model of Emissions of Gases and Aerosols from Nature (MEGAN; Guenther et al., 2006) in both domains. These are complemented by the Emissions Database for Global Atmospheric Research Hemispheric Transport of Air Pollution (EDGAR-HTAP; Janssens-Maenhout et al., 2012) anthropogenic emissions for the Southern African domain, while using the 2017 emission inventory prepared by the US Environmental Protection Agency (EPA; https://www.epa.gov/air-emissions-inventories/2017-national-emissions-inventory-nei-data; last access: 21 April 2022) for

North America. Naturally, VFEI was used in both simulations to account for BB emissions.

Panels a and b from figure 5 show the total carbon emissions calculated by VFEI in each domain during their corresponding simulation periods. It is evident that fire activity is more widespread in the African simulation and with higher magnitudes. In areas between 5° S and 20° S the carbon emissions easily exceed 0.2 Tg, with local areas in Mozambique and Zambia surpassing 10 Tg. On the other hand, in North America fires are not widespread due to their different nature mentioned above.

Thus, total carbon amounts from BB are much smaller than the ones in the African domain. Here, the areas with higher emissions are concentrated in the Pacific Norwest region, precisely the focus area of FIREX-AQ during this period. In fact, a large fire event occurred in early August 2019, and this is detected by VFEI and by the other four inventories as well (Fig. 5d). It is important to mention that VFEI carbon emissions seem to be lower than the other inventories in both simulations, especially in North America. This can happen since, in contrast to the other inventories, VFEI does not include a treatment for

fires potentially obscured by clouds. This phenomenon can be particularly important in fires occurring in mid and high latitudes because the periodic passage of frontal systems and other precipitation events. Nonetheless, a future version of VFEI will include a correction for this following the same approach as in QFED (see section 6).





In Southern Africa, BB emissions where distributed daily by using a gaussian distribution with a peak at 2 PM local solar time as suggested in Roberts et al. (2009). In North America, FRP data from the active fire product of the Geostationary Operational

Environmental Satellites (GOES-16 and GOES-17; Li et al., 2020) was used to constrain the BB diurnal cycle. However, most observations from these datasets are from large fires only. For other fires the BB diurnal cycle was constrained depending on biome type following Li et al. (2019). It is important to note that the FRP from the GOES satellites was used only to constrain the diurnal cycle and not the emissions.

BB emissions were also distributed vertically using the plume rise model described by Freitas et al. (2007, 2010), which uses

the meteorological conditions from WRF-Chem to compute the injection layer of smoke. We used this process only for the North American simulation. Ferrada et al. (2021) showed that the use of plume rise in Southern Africa could lead to overestimations of smoke in the mid-troposphere and found better results when emissions were distributed within the boundary layer.

### 4.3 AOD evaluation

In this section we evaluate modeled AOD against observations from MODIS MAIAC (Lyapustin et al., 2018). MODIS MAIAC provides high quality AOD retrievals at 1 km resolution globally. In this study, we used preliminary results based on the MODIS MAIAC collection 6.1 (C6.1). This new version provides several improvements from the collection 6 (C6). The main one is the use of updated regional aerosol models correcting for the known biomass burning AOD underestimation at high AOD values (AOD>0.6). The updated models, derived from analysis of the v3 AERONET dataset, have a dynamically

increasing absorption and reduced effective particle size with AOD at AOD>0.6, typical of the emitted BB smoke in comparison to the background aerosol (Lyapustin et al., *manuscript in preparation*). MAIAC 6.1 also introduced a 300 km buffer zone covering the transition between different MAIAC regions with different aerosol model absorption. The linear change of the aerosol model properties from one region to another avoids AOD discontinuity on the regions' boundary. We gridded MAIAC C6.1 1 km resolution data into each model grid by taking the average of all observations falling within a grid

box. In addition, we used the AERONET sun photometer data (version 3, level 2.0, cloud-cleared and quality assured; Giles et al., 2019) from available stations in each domain. The data from the model simulations were extracted by using the closer grid cell to each AERONET site.

From the model side, the optical properties of each aerosol bin included in MOSAIC are calculated using the Mie code (Toon and Ackerman, 1981). Then, the bulk optical properties of the aerosols are calculated by simply summing across all bins (Fast

et al., 2006; Barnard et al., 2010). The Mie code provides aerosol optical properties in 4 wavelengths (300, 400, 600 and 1000 nm). Here, we compare AOD against MODIS MAIAC and AERONET sites at wavelengths of 550 and 500 nm, respectively. We use the model data from wavelengths 400 and 600 nm to calculate the Angstrom exponent ($\alpha$) to then estimate the AOD at the desired wavelength ($\lambda$), as follows:

$$\alpha = -\frac{\log\left(\frac{AOD_{400}}{AOD_{600}}\right)}{\log\left(\frac{400}{600}\right)}, \tag{4}$$



$$AOD_\lambda = AOD_{400} \left(\frac{\lambda}{400}\right)^{-\alpha}$$
(5)

### 4.3.1 Southern Africa

Figure 6 shows a composite of averaged AOD at 550 nm during September 2016 over Southern Africa from MODIS MAIAC C6.1, the WRF-VFEI simulation and two reanalysis products: CAMS and the Modern-Era Retrospective analysis for Research and Applications (MERRA-2; Gelaro et al., 2017). The data from the model and reanalysis products were sampled at the

locations and observation times of MODIS MAIAC, whenever there was observation. Thus, this is a direct comparison of each dataset against observations. The number of observations (Fig. 6e) gives an insight of how robust these averages in each grid point are. Evidently, land areas with little or absent cloud coverage throughout the period are the ones with higher number of observations and, thus, samples from the models. Observed AOD suggests that the smoke plume was located mostly over Angola, western Zambia, and the southern regions of the Democratic Republic of the Congo (DRC) with magnitudes between

0.4 and 0.8 (Fig. 6a). Although, the location of the core of the smoke plume is well reproduced by WRF-VFEI and the reanalysis products (Fig. 6b-d), they differ on their magnitudes. WRF-VFEI shows overestimations in AOD of around 0.14 in the western side of Southern Africa, while slightly underestimates the AOD in the east side. The higher biases appear in northwest Angola and the westernmost tip of the DRC, however, a relatively few number of observations is weighted in the average in these areas. In general, the reanalysis products underrepresent the AODs by around 0.1 in the case of CAMS and

0.22 in the case of MERRA-2.

Figure S2 includes the same comparison as in figure 6 but using MAIAC C6 instead. In general, the AOD in the source region is higher in C6.1 than in C6, due to the improved representation of the aerosol properties in high AOD regions. Additionally, the boundary between two different aerosol models is noticed at around 18° S (Fig. S2). As a result, AOD south of this parallel is lower compared to the northern side, leading to noticeable biases from WRF-VFEI and reanalyses. In contrast, this does not

occur when using MAIAC C6.1 (Fig. 6).

To make a more extensive analysis, we compare WRF-VFEI against AERONET data (Fig. 7). We selected six sites that were closer to the core of the smoke plume to have a better insight on the inter-diurnal variability of AOD (Fig. 7a-f). In general, WRF-VFEI is able to reproduce the trends with accuracy, with correlations of ~0.7 and mean AOD biases of around 0.1. We highlight the stations of Windpoort and Bonanza (Fig. 7e,f), both located in Namibia within the region with different aerosol

model in MAIAC as discussed above. WRF-VFEI not only reproduces the AOD variability very well in both stations (correlations higher than 0.86), but the AOD biases are also small (lower than 0.1).

During September 2016, 24 AERONET sites had data for comparison. When considering all this information available (more than 36000 data points) we find that WRF-VFEI has an overall mean AOD bias of +0.029 and correlation of 0.85 (Fig. 7h). These metrics confirm a good agreement between VFEI emissions and observations.





### 4.3.2 North America


We applied the same approach to evaluate the simulation over North America. Figure 8 shows the comparison of MODIS MAIAC AOD against WRF-VFEI and the two reanalysis products included as reference: CAMS and MERRA-2. Overall, AODs in this domain are much smaller than in Southern Africa since North America does not experience widespread fire activity. In contrast, BB tends to be mostly controlled fires, with occasional wildfires that often ignite in the west side of the

continent. Moreover, sources of aerosols in North America are more diverse, with anthropogenic aerosols playing and important role in the AOD. In addition, sporadic long-range transport of smoke can contribute to high aerosol loadings. With this in consideration, WRF-VFEI and the reanalysis products capture the main features of the AOD pattern in North America (Fig. 8b), with the higher AODs (> 0.2) in the densely populated areas of the Eastern and Southern US, which is also observed by MODIS MAIAC and the reanalysis products. Furthermore, WRF-VFEI reproduces AODs higher than 0.3 in Central Canada

that were produced by a relatively intense fire activity in that region during this period. Overall, WRF-VFEI seems to slightly underestimate AODs in the domain (biases of -0.04).

It is important to note that these AODs consider all sources of aerosols (BB, anthropogenic and biogenic). Since large fires in North America tend to be more sporadic, we focused on the largest fire event observed during this period which occurred in early August 2019 in the state of Washington in the US (47.97° N, 118.45° W). This fire was called as Williams Flats fire

(WFF). Figure 9 shows a similar composite as figure 8, but considering the nested domain used in this simulation and averaged only between 6-9 August 2019 to isolate this fire event. Here, the smoke from the fire is clearly visible in MODIS MAIAC with AODs ranging from 0.3 to 0.7 during this period. The averaged map also captures the long-range transport of the smoke over Northern Idaho and Montana. WRF-VFEI reproduces the smoke from this fire well, although with smaller AODs over Montana (Fig. 9f). This can be due to several reasons: the resolution of the inner domain (6.7 km) is not high enough to

reproduce abrupt changes in topography that are common in this mountainous region; the plume rise process may be injecting smoke at altitudes not high enough for it to be subject to stronger winds, which has been documented in Ye et al. (2021); and/or chosen parameterizations have their own uncertainties that contribute to errors in the meteorology and subsequent transport of the smoke. The reanalysis products show similar patterns as observed by MODIS MAIAC with local biases, especially in the fire source. However, the details of the plume are difficult to evaluate in these reanalysis products since they provide data at

coarse resolutions (CAMS at ~80 km and MERRA-2 at ~25 km). For visualization purposes, these datasets were gridded into the model grid.

Figure 10 shows an AOD comparison between AERONET and WRF-VFEI. Six stations were chosen to be analyzed as time series. Five of these stations were the closest to the WFF (Fig. 10a-e) while the UWisconsin_SSEC site (Fig. 10f) was chosen because it was affected by long-range transport of smoke from Canada in late July 2019. Hence, the data from these stations

can be considered to contain a strong signal from BB rather than aerosols from other emission sources. In general, WRF-VFEI reproduces the trends and day-to-day variability in these stations with correlations above 0.9 in UWisconsin_SSEC and 0.79 in Missoula. In contrast, correlations were low (< 0.4) in Cliff_Creek_6, probably due to its location in highly complex terrain



(mountainous region). Another feature observed is that WRF-VFEI did not always reproduce the timing of the smoke plume over each station. For example, on 9 August the Rimrock AERONET site (Fig. 10d) experienced high AODs that decreased quickly (from ~0.33 to 0.17 in a matter of few hours). WRF-VFEI also reproduced the same trend but around 4 hours in advance. These situations obviously have an impact on the metrics used to evaluate the performance of the model and highlights the need for multiple observations of fires throughout the day to be able to parametrize better the diurnal cycle of BB emissions. Overall, when considering all 109 AERONET stations available within the domain, results show that the model is capable of reproducing AOD values with high accuracy (mean bias of -0.013 and correlation of 0.73; Fig. 10h). However, as mentioned above, the AODs in this domain are not always related to BB, but in many cases due to anthropogenic emissions (urban areas, transport, industries, etc.).

### 4.4 Comparison against airborne data

#### 4.4.1 ORACLES

The ORACLES field campaign provided airborne observations of smoke from Southern Africa over the South Atlantic Ocean during the three years the mission was deployed: September 2016, August 2017, and October 2018. Here, we use in-situ observations of CO from the CO Measurement & Analysis (COMA) trace gas detector and black carbon concentrations from the Single Particle Soot Photometer (SP2) onboard the P-3 aircraft (12 flights during September 2016). As described in Redemann et al. (2020), these flights were conducted at different altitudes with different maneuvers and mostly at altitudes of around 5-6 km.a.s.l (Fig. S6). To make a fair comparison, we averaged the data every 60 seconds to remove its noise. Even though no flights were conducted over land or near the source of fires, we assume that these species suffer little alteration throughout their westward transport over the ocean.

Overall, WRF-VFEI reproduces these species accurately (Fig. 11), with correlations higher than 0.77 and normalized mean biases of around –2 % for BC. WRF-VFEI tends to underestimate CO more considerably (~30 ppb$_v$) but consistently throughout all flights and altitudes (Fig. S7). This may suggest that the background CO in the model is underestimated, rather than VFEI emissions themselves. However, this is difficult to prove due to lack of observational data outside the smoke plume. Also, it is important to consider that most observations were made at altitudes higher than 3 km where the model layers are thicker (~800 m) and we used the closest altitude approach to extract the data from the model, which can also introduce errors. Furthermore, the SP2 instrument has a nominal uncertainty of 25 % of the BC reported and, when considering this, most of the BC from the model falls under the uncertainty range of the instrument.

#### 4.4.2 FIREX-AQ

Analogously, the FIREX-AQ campaign measured diverse trace gases with different instruments. Here, we focused on CO, formaldehyde, methanol, SO$_2$ and BC (Fig. 12). Using the same approach applied to the ORACLES airborne data, we averaged the data every 60 seconds to remove possible noise and abrupt changes in the concentration of these species. Figure S8 shows



the flight trajectories of each of the 9 flights from the campaign that covered the period of the model simulation. Here, WRF-
VFEI produces accurate results when comparing CO concentrations of less than 150 ppb, but biases are greater at higher
concentrations (Fig. 12a). The same trend is observed for the other species, with overall underestimations. This can be
happening for one or a combination of the following reasons: (i) VFEI slightly underestimates the BB emissions over the North
American Pacific Northwest during this period. Figure 6d hints that VFEI has overall lower carbon emissions than other
inventories such as QFED and GFAS during the simulation for FIREX-AQ. However, VFEI emissions are comparable to these
inventories during the 2012-2019 period (Fig. 4c); (ii) generally, fires in North America tend to be localized and thus, the exact
timing, location, evolution, and transport of the smoke is more challenging for the model to reproduce, especially at the
resolutions used in this study (20 km and 6.7 km for the outer and inner domains, respectively, with 52 vertical layers). Since
the comparison against the flights was done using the nearest grid point to the trajectories of the flights, even a slight
misrepresentation of the smoke can introduce large biases in the results. For example, the model can be simulating the plume
transport faster or slower than observed; or the core of the plume can be displaced more to the north or south. (iii) in contrast
to ORACLES, the flights conducted during FIREX-AQ flew directly into the smoke plumes with zigzag transects going rapidly
in and out of the plume covering relatively small areas (Fig. S8). Because of this, it is not surprising that the model could not
reproduce these drastic changes in CO (Fig. S10) and other species, since the data was extracted from few grid boxes. It is
important to remember that the model reports the average concentrations in each grid box, hence, hiding extreme values and
smoothing sharp gradients. This is especially the case of the flights of 6, 7 and 8 August 2019 which sampled the smoke from
the Williams Flats fire (Fig. S10g-i), where the model is not able to reproduce the drastic changes in CO due to one or more
of the factors mentioned.

**5 Resolution advantage of VFEI**

In section 4, we evaluated the quantities of VFEI emissions in two mid-resolution model simulations. Nonetheless, another
capability of VFEI is its high-resolution (~500 m) that can improve the placement of fires in fine resolution model grids. This
is on the same direction as other BB inventories, such as FEER and FLAMBE, which provide data at MODIS pixel resolution
(1 km nadir) instead of gridding the data at coarse resolutions. Figure 13 shows an example of this advantage by illustrating
the evolution of the WFF during early August 2019 and how emissions are mapped into a high-resolution (3 km) model grid.
If VFEI would have a typical spatial resolution of 0.1° (~10 km; Fig. 13a), this single large fire is then mapped as 1-5 different
isolated fires into the 3 km model grid (Fig. 13b). This happens because models assign each data point from the inventory into
the closer coordinates of the model grid. This can be particularly important in fires over complex topography since irregular
terrain can promote different local circulation patterns. Thus, altering model processes like plume rise that uses the local
meteorological profile (temperature, humidity, winds, etc.) of the grid box where the fire is located to distribute the emissions
vertically (Freitas et al., 2007, 2010) and, therefore, altering the transport of the smoke plume. In contrast, the high resolution
of VFEI (~500 m) is capable to reproduce the day-to-day evolution of the fire and its area more accurately (Fig. 13c). Thus,





when mapping this information into the 3 km model grid (Fig. 13d), the model is capable to constraint the fire and its area more precisely, and thus, minimizing the possible smoke transport complications mentioned before.

The potential misrepresentation of vertical distribution of smoke and its latter transport not only applies to large fire events. In addition, small fires, such as agricultural prescribed fires, are also prone to this, because small fires can be notoriously
misplaced during the mapping process. Expanding the example above, that would mean that a small fire of an area of 1 ha would have a location uncertainty radius of up to around 0.1° (~10 km) if using the 0.1° resolution inventory, while if using VFEI at its native resolution, that uncertainty is reduced to ~500 m or less.

In this study, the high-resolution advantage of VFEI was not evaluated because of the high computational cost that high-resolution model simulations involve. Consequently, just the quantities of the emissions were evaluated in section 4. However,
as computational resources keep increasing, it is expected that very fine resolution model simulations will become more common and will be able to take advantage of this capability of VFEI.

## 6 Summary and conclusions

We developed an open BB emission inventory called VIIRS-based Fire Emission Inventory (VFEI). VFEI produces daily emissions for 46 species of aerosols and gases by using the FRP data from the VIIRS instrument onboard the Suomi-NPP
satellite (*top-down* approach). The formula is the same as the one used in other inventories, such as QFED and GFAS. Emissions for each species depend on the FRP density, an empirical factor to relate FRP to the amount of biomass above the ground and the emission factor of the specie. VFEI uses the emission factors presented by Andreae (2019) distinguishing among 6 different biomes. The main advantages of using VIIRS instead of MODIS is its ability to detect small fires, absence of data gaps over the tropics and its higher resolution (375 m on its I-Band). Thus, VFEI provides emissions at ~500 m
resolutions which is ideal for high-resolution model simulations. In addition, the data product comes in two different formats for an easy implementation in models (as text or NetCDF files). Globally, VFEI produces carbon emissions comparable to other major inventories, such as GFED, QFED and GFAS, but with some differences depending on the region of the globe. We note that this initial version of VFEI (VFEIv0) does not include a treatment for fires that can be missed due to cloud coverage, which can explain the overall slightly lower emissions in high latitudes when compared against the other inventories.
We anticipate that the next version of VFEI (VFEIv1) will include the same approach used in QFED and described by Darmenov and da Silva (2013) to solve this potential problem. Future versions will also include observations from the VIIRS instrument onboard the NOAA-20 satellite when its fire product becomes publicly available.

Two simulations using the WRF-Chem model and VFEI were conducted to assess the BB emissions: the first over Southern Africa during September 2016, and the second over North America during July-August 2019. These regions and periods were
chosen to make use of two datasets from different field experiments (ORACLES and FIREX-AQ, respectively). Results showed that VFEI produces strong correlations (0.85) of AOD when compared against MODIS MAIAC C6.1 and AERONET stations over Southern Africa. In North America correlations were also high (0.73), but in this region is more difficult to make





a clear assesment because of the different sources of aerosols. For example, anthropogenic emissions play an important role. By isolating the case for the largest fire ocurred during the simulation (Williams Flats fire), AOD results showed overall good

agreement with the MODIS MAIAC C6.1 observations, but with slight discrepancies on the exact location and AOD magnitudes of the smoke plume. When comparing against the flight data from each field campaign, the model performed well over Southern Africa, with CO correlations of 0.78 and slight BC underestimations of around 2 %. However, the model reproduces the smoke plumes sampled by FIREX-AQ with low correlations and underestimating high concentrations of CO. This is because the trajectories of the flights conducted during FIREX-AQ were performed in transects coming in and out of

the plumes rapidly and the model could not reproduce the exact location and timing of the smoke plumes, thus producing large biases (other possible factors were exposed in section 4.4.2). On the other hand, the good results over Southern Africa can be explained because the widespread and persistent fire activity of the region that produces an almost homogeneous smoke plume that the model is able to reproduce more accurately. It should be noted that we only conducted two regional model simulations for a month each, and thus, results are particular to these regions and periods. Ideally, global model simulations for at least a

characteristic year would be necessary to widely assess VFEI and, perhaps, the discussion for possible callibrations.

It is necessary to mention that VFEI, as the other inventories, provides an estimation of BB emissions. The formulation used to derive emissions use different factors that have been generalized. For instance, emission factors are a compilation of several in-situ and laboratory studies, but not all species have been extensively sampled, nor there is enough data for different biomes. In addition, the FRP data from VIIRS also introduces uncertainties that are inherent to the active fire product and the instrument

itself. Furthermore, BB inventories still do not provide differentiation between flaming or smoldering phases, which can impact the calculated emission fluxes. New developments on remote sensing techniques can provide an insight on this by deriving the visible fire energy which is strongly correlated to the modified combustion efficiency (Wang et al., 2020), which could significantly improve the emissions and information reported by VFEI. Another aspect that can be improved by remote sensing studies is the diurnal cycle used in models to release the BB. Commonly, a gaussian distribution with a peak at around 2 PM

local time is used, but some fires (especially large fire events) do not follow this cycle, leading to errors in the model results.

It is important to reflect that atmospheric models have been developed in recent decades to consider complex chemical reactions and aerosol formation and feedbacks. Nonetheless, they are often optimized for some regions of the globe. Furthermore, an important source of uncertainty is the vertical distribution of smoke, which is often done by using a plume rise model. The dynamical approach used in WRF-Chem uses tabulated data of heat fluxes from Amazon fires, which can

introduce errors when used in other regions. Improved versions of plume rise could provide a more individual treatment to better distribute emissions within the vertical layers of the model and, thus, improving the smoke transport. Another aspect is the calculation of AOD and aerosol properties, which can vary from model to model. Recent studies (e.g., Majdzadeh et al., 2022), have discussed and proposed new approaches to calculate aerosol properties which could improve the accuracy of the reported AODs from the model.




**Code and data availability:** The VFEIv0 product containing daily emission fluxes from early 2012 to 2021 can be accessed at https://doi.org/10.5281/zenodo.6474058 (doi:10.5281/zenodo.6474058). The data can also be accessed through http://bio.cgrer.uiowa.edu/VFEI/DOWNLOAD (last access: 21 April 2022), which also contains near-real-time emissions. The VIIRS I-band active fire product is provided through the Level-1 and Atmosphere Archive & Distribution System

(LAADS) Distributed Active Archive Center (DAAC) website (https://ladsweb.modaps.eosdis.nasa.gov/archive/; last access: 21 April 2022). The MODIS IGBP data was downloaded from the Land Processes Distributed Active Archive Center (LP DAAC) of the United States Geological Survey (USGS) website (https://lpdaac.usgs.gov/products/mcd12c1v006/; last access: 21 April 2022). The GFED emission datasets were downloaded from their main website (https://globalfiredata.org/pages/data/; last access: 21 April 2022). Similarly, FINN emissions were downloaded from the NCAR's Atmospheric Chemistry

Observations and Modeling (ACOM) repository (https://www.acom.ucar.edu/Data/fire/; last access: 21 April 2022). QFED daily emissions were obtained from NASA's Center for Climate Simulation (NCCS) public repository (https://portal.nccs.nasa.gov/datashare/iesa/

aerosol/emissions/QFED/v2.5r1/0.1/QFED/; last access: 21 April 2022). GFAS emissions were retrieved from ECMWF's public online repository (https://apps.ecmwf.int/datasets/data/cams-gfas/; last access: 21 April 2022). The WRF-Chem model

v4.2 was downloaded from their public GitHub repository (https://github.com/wrf-model/WRF/releases/tag/v4.2; last access: 21 April 2022). The MODIS MAIAC product was retrieved from USGS' LP DAAC website (https://lpdaac.usgs.gov/products/mcd19a2v006/; last access: 21 April 2022). The preliminary MAIAC C6.1 dataset was provided by the MAIAC team via the NASA Computer for Climate Simulations (NCCS). The AERONET data can be accessed through their dedicated website (https://aeronet.gsfc.nasa.gov/cgi-bin/webtool_aod_v3; last access: 21 April 2022). The

airborne data from the ORACLES campaign was retrieved through NASA's Earth Science Project Office (ESPO) website (https://espo.nasa.gov/oracles/archive/browse/oracles; last access: 21 April 2022), while the data from the FIREX-AQ field experiment can be accessed through NASA's Langley Research Center data archive (https://www-air.larc.nasa.gov/cgi-bin/ArcView/firexaq; last access: 21 April 2022).

**Author contributions:** GAF was responsible for the design, methodology, development, and coding of VFEI, model simulations, analysis of results and writing of the most part of the manuscript. GRC and SRF provided comments on the research design, modeling details and interpretation of results. MZ and JW processed, filtered, and gridded the MODIS MAIAC C6 and C6.1 data that was used to evaluate model results. Additionally, MZ provided comments on the methods of VFEI. AL and YW provided the newer MODIS MAIAC C6.1 version details and data used to evaluate the model simulations for this

study. All authors provided comments and edited the manuscript.

**Competing interests:** The authors declare that they have no conflict of interest.



**Acknowledgements:** The authors acknowledge the Level-1 and Atmosphere Archive & Distribution System (LAADS)
Distributed Active Archive Center (DAAC), located in the Goddard Space Flight Center in Greenbelt, Maryland
(https://ladsweb.nascom.nasa.gov/; last access 01 February 2022) from where the VIIRS active fire data was retrieved. We
would also like to acknowledge high-performance computing support from Cheyenne (doi:10.5065/D6RX99HX) provided by
NCAR's Computational and Information Systems Laboratory, sponsored by the National Science Foundation, where the model
simulations from this study were performed. The authors thank the PIs in charge of the instruments deployed in the ORACLES
and FIREX-AQ field experiments that measured the species used to evaluate the model simulations in this study: James
Podolske (CO; ORACLES); Steven Howell and Steffen Freitag (BC; ORACLES); Jeff Peischl and Tom Ryerson (CO; FIREX-
AQ); Reem A. Hannun and Thomas F. Hanisco (formaldehyde; FIREX-AQ); Carsten Warneke, Matthew Coggon, Georgios
Gkatzelis and Kanako Sekimoto (methanol; FIREX-AQ); Andrew W. Rollins ($SO_2$; FIREX-AQ) and Joshua P. Schwarz (BC;
FIREX-AQ). Special thanks to Dr. Ravan Ahmadov (CIRES/NOAA) who provided advise during the making of this study.
We further acknowledge the use of imagery from the Worldview Snapshots application (https://wvs.earthdata.nasa.gov), part
of the Earth Observing System Data and Information System (EOSDIS).

**Financial support**: This research has been supported by the National Aeronautics and Space Administration (NASA; grant
no. NNX15AF95G) and the National Oceanic and Atmospheric Administration (grant no. NA16OAR4310114). M. Zhou and
J. Wang also acknowledge the support from NASA (grant no. 80NNSC21L1976 and 80NSSC21K1494). The work of A.
Lyapustin and Y. Wang was supported by the NASA MODIS Terra/Aqua and VIIRS SNPP Program.





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





**Table 1.** Species considered by VFEI and their emission factors (g kg$^{-1}$) compiled by Andreae (2019).

| Species | VFEI name | Tropical forest | Temperate forest | Boreal forest | Savanna & grassland | Cropland | Peat fires |
|---|---|---|---|---|---|---|---|
| CO | CO | 104 | 113 | 121 | 69 | 76 | 260 |
| $CO_2$ | CO2 | 1620 | 1570 | 1530 | 1660 | 1430 | 1590 |
| $CH_4$ | CH4 | 6.5 | 5.2 | 5.5 | 2.7 | 5.7 | 9.1 |
| $C_2H_2$ | C2H2 | 0.35 | 0.31 | 0.28 | 0.31 | 0.27 | 0.11 |
| Ethylene | C2H4 | 1.11 | 1.11 | 1.54 | 0.83 | 1 | 1.47 |
| Ethane | C2H6 | 0.88 | 0.69 | 0.97 | 0.42 | 0.79 | 1.85 |
| $C_3H_4$ | C3H4 | 0.013 | 0.05 | 0.062 | 0.071 | 0.18 | 0.006 |
| Propylene | C3H6 | 0.86 | 0.6 | 0.67 | 0.46 | 0.47 | 1.14 |
| Propane | C3H8 | 0.53 | 0.28 | 0.29 | 0.13 | 0.17 | 0.99 |
| Butenes | BUTENES | 0.246 | 0.281 | 0.265 | 0.16 | 0.225 | 0.91 |
| Butanes | BUTANES | 0.056 | 0.111 | 0.163 | 0.028 | 0.059 | 0.41 |
| Isoprene | ISOP | 0.22 | 0.1 | 0.074 | 0.101 | 0.17 | 0.52 |
| Pentenes | PENTENES | 0.084 | 0.091 | 0.057 | 0.036 | 0.038 | 0.172 |
| Pentanes | PENTANES | 0.022 | 0.051 | 0.082 | 0.015 | 0.061 | 0.363 |
| Hexenes | HEXENES | 0.151 | 0.182 | 0.195 | 0.11 | 0.042 | 0.34 |
| Hexanes | HEXANES | 0.08 | 0.058 | 0.067 | 0.037 | 0.099 | 0.194 |
| Heptanes | HEPTANES | 0.024 | 0.029 | 0.021 | 0.016 | 0.031 | 0.112 |
| Octenes | OCTENES | 0.012 | 0.036 | 0.021 | 0.021 | 0.002 | 0.065 |
| Terpenes | TERPENES | 0.15 | 1.17 | 1.53 | 0.104 | 0.027 | 0.08 |
| Benzene | BENZENE | 0.38 | 0.42 | 0.57 | 0.33 | 0.27 | 0.87 |
| Toluene | TOLUENE | 0.23 | 0.27 | 0.35 | 0.19 | 0.17 | 0.45 |
| Xylenes | XYLENES | 0.086 | 0.16 | 0.11 | 0.086 | 0.1 | 0.23 |
| Methanol | METHANOL | 2.8 | 2.2 | 2.33 | 1.35 | 3.3 | 2.5 |
| Ethanol | ETHANOL | 0.067 | 0.076 | 0.058 | 0.036 | 0.05 | 0.17 |
| Formaldehyde | HCHO | 2.4 | 2.04 | 1.75 | 1.23 | 1.8 | 1.07 |
| Acetaldehyde | CH3CHO | 2.26 | 1.21 | 0.81 | 0.84 | 1.8 | 1.16 |
| Glyoxal | GLYOXAL | 0.5 | 0.54 | 0.59 | 0.33 | 0.24 | 1.3 |
| Methylglyoxal | CH3COCHO | 0.49 | 0.27 | 0.57 | 0.4 | 0.55 | 0.23 |
| Methacrolein | MACR | 0.15 | 0.14 | 0.11 | 0.1 | 0.28 | 0.39 |
| Acetone | ACETONE | 0.63 | 0.76 | 1.59 | 0.47 | 0.71 | 0.91 |
| Methyl vinyl ketone | MVK | 0.39 | 0.165 | 0.099 | 0.23 | 0.48 | 0.057 |
| Acetol | HYAC | 1.81 | 1.13 | 2.1 | 0.56 | 3.12 | 0.64 |
| Formic acid | HCOOH | 0.49 | 0.91 | 1.04 | 0.21 | 0.56 | 0.29 |
| Acetic acid | CH3COOH | 3.3 | 2.74 | 3.8 | 2.31 | 6.1 | 4.9 |
| $H_2$ | H2 | 3.1 | 2.1 | 1.6 | 0.97 | 2.6 | 1.2 |
| NO | NO | 2.8 | 3 | 1.18 | 2.5 | 2.4 | 1.2 |
| HONO | HONO | 0.85 | 0.33 | 0.41 | 0.47 | 0.37 | 0.35 |
| $N_2O$ | N2O | 0.2 | 0.25 | 0.24 | 0.17 | 0.09 | 0.17 |
| $NH_3$ | NH3 | 1.33 | 0.98 | 2.5 | 0.89 | 0.99 | 4.2 |
| $SO_2$ | SO2 | 0.77 | 0.7 | 0.75 | 0.47 | 0.8 | 4.3 |
| Dimethyl sulfide | DMS | 0.0022 | 0.014 | 0.0023 | 0.008 | 0.05 | 0.045 |
| HCl | HCL | 0.13 | 0.039 | 0.13 | 0.13 | 0.18 | 0.008 |
| $PM_{2.5}$ | PM25 | 8.3 | 18.5 | 18.7 | 6.7 | 8.2 | 18.9 |
| Total PM | TPM | 10.9 | 18.4 | 15.3 | 8.7 | 12.9 | 27.5 |
| Organic carbon | OC | 4.4 | 10.9 | 5.9 | 3 | 4.9 | 14.2 |
| Black carbon | BC | 0.51 | 0.55 | 0.43 | 0.53 | 0.42 | 0.1 |

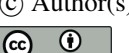



**Table 2.** Biomes considered in VFEI with their corresponding mapping using IGBP land classes and Köppen climate classifications. See footnote for the key of each climates.

| VFEI biome | IGBP land classification | Köppen climate classifications |
|---|---|---|
| Tropical forest | Evergreen Broadleaf Forest | Af, Am, Aw, BSh, Cwa, Cwb |
| Tropical savanna | Closed Shrublands | Af, Am, Aw, BSh, Cwa, Cwb |
| | Open Shrublands | |
| | Woody Savannas | |
| | Savannas | |
| Temperate forest | Evergreen Needleleaf Forest | Climates that do not belong to tropical nor boreal biomes |
| | Evergreen Broadleaf Forest | |
| | Deciduous Needleleaf Forest | |
| | Deciduous Broadleaf Forest | |
| | Mixed Forest | |
| Temperate savanna | Closed Shrublands | Climates that do not belong to tropical nor boreal biomes |
| | Open Shrublands | |
| | Woody Savannas | |
| | Savannas | |
| Boreal forest | Evergreen Needleleaf Forest | Dsc, Dsd, Dwc, Dwd, Dfc, Dfd, ET, EF |
| | Deciduous Needleleaf Forest | |
| | Mixed Forest | |
| Boreal savanna | Closed Shrublands | Dsc, Dsd, Dwc, Dwd, Dfc, Dfd, ET, EF |
| | Open Shrublands | |
| | Woody Savannas | |
| | Savannas | |
| Grassland | Grasslands | Climate independent |
| | Urban and Built-Up | |
| Cropland | Croplands | Climate independent |
| | Cropland/Natural Vegetation Mosaic | |
| Peatland | Permanent Wetlands | Climate independent |
| Desertic areas | Snow and Ice | Climate independent |
| | Barren or Sparsely Vegetated | |

| | |
|---|---|
| (Af) Tropical rainforest | (Dsd) Cold, dry summer, very cold winter |
| (Am) Tropical monsoon | (Dwc) Cold, dry winter, cold summer |
| (Aw) Tropical savanna | (Dwd) Cold, dry winter, very cold winter |
| (BSh) Arid, hot steppe | (Dfc) Cold, no dry season, cold summer |
| (Cwa) Temperate, dry winter, hot summer | (Dfd) Cold, no dry season, very cold winter |
| (Cwb) Temperate, dry winter, warm summer | (ET) Polar, tundra |
| (Dsc) Cold, dry summer, cold summer | (EF) Polar, frost |






**Table 3.** Regions defined for analysis.

| Region | Abbreviation | Bounding box | | | |
|---|---|---|---|---|---|
| | | West | East | South | North |
| Boreal America | BAm | 168° W | 55° W | 49° N | 72° N |
| North America | NAm | 126° W | 60° W | 25° N | 49° N |
| Central America | CAm | 110° W | 50° W | 0° | 25° N |
| South America | SAm | 82° W | 30° W | 30° S | 0° |
| Europe | Eur | 12° W | 50° E | 35° N | 73° N |
| Northern Africa | NAf | 20° W | 45° E | 0° | 25° N |
| Southern Africa | SAf | 8° E | 51° E | 35° S | 0° |
| Siberia | Sib | 85° E | 150° E | 42° N | 75° N |
| Southeast Asia | SEA | 70° E | 130° E | 0° | 28° N |
| Australasia | AUS | 95° E | 160° E | 45° S | 0° |



**Table 4.** Details of inventories used to compare VFEI against.

| Parameter | Inventory | | | |
|---|---|---|---|---|
| | GFED | FINN | QFED | GFAS |
| Approach | bottom-up | bottom-up | top-down | top-down |
| Hotspots | MODIS burnt area | MODIS burnt area | MODIS active fire FRP | MODIS and SEVIRI FRP |
| Land cover | MODIS | MODIS IGBP | MODIS INPE-IGBP | MODIS |
| Emission factors | Andreae & Merlet (2001) | Akagi et al. (2011) | Andreae & Merlet (2001) | Akagi et al. (2011) |
| | | Andreae & Merlet (2001) | | |
| | | McMeeking (2008) | | |
| Product resolution | 0.25 degrees | ~1 km | 0.1 degrees | 0.1 degrees |
| Reference(s) | van der Werf et al. (2010, 2017) | Wiedinmyer et al. (2011) | Darmenov & da Silva (2015) | Kaiser et al. (2012) |

INPE: Instituto Nacional de Pesquisas Espaciais (Portuguese); *National Institute for Space Research (English).*





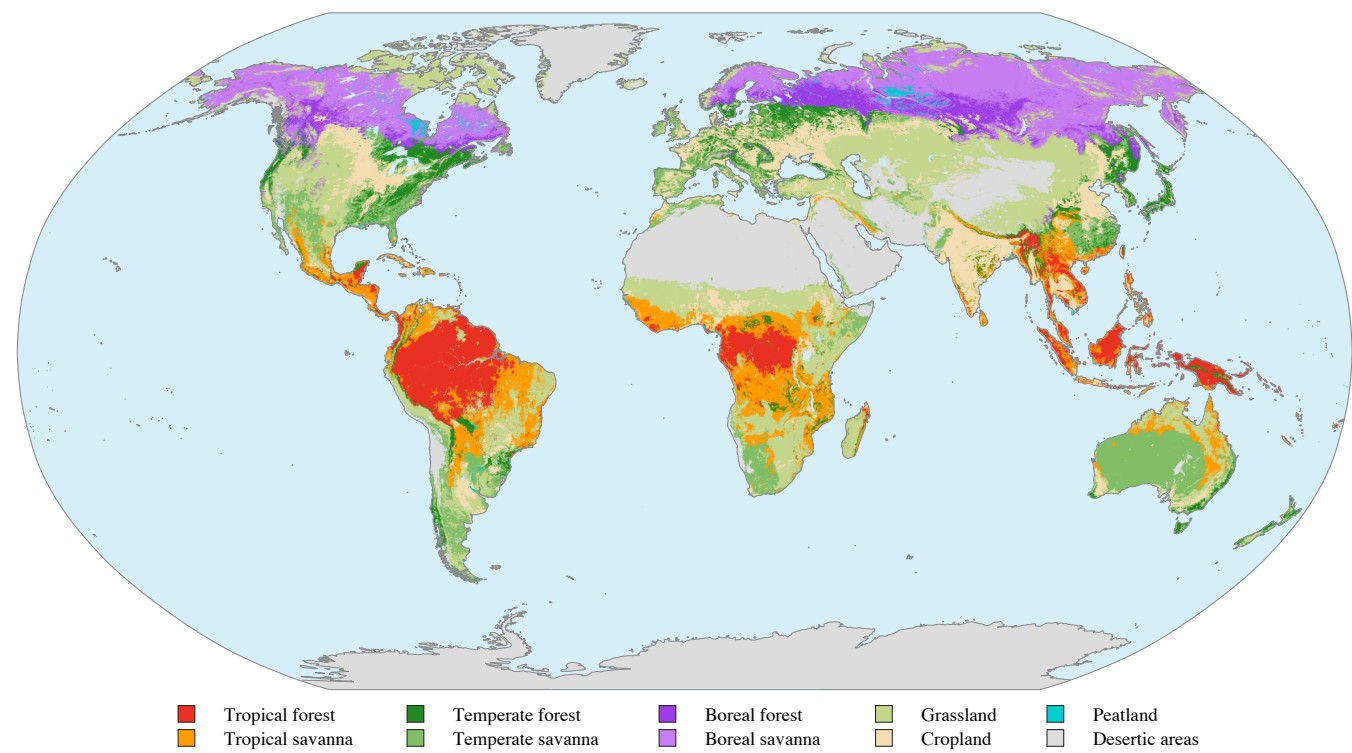

**Figure 1.** Biome map used in VFEI. Biomes were constrained using MODIS IGBP land categories and Köppen climate classifications following the criteria shown in table 2.





(a)

(b)

**Figure 2.** (a) Average annual carbon emissions (g m⁻² a⁻¹) from VFEI between 2012 and 2019. Polygons show the regions defined to compare VFEI to other inventories. (b) Annual cycle of carbon emissions in each region (Tg month⁻¹) defined in table 3 and shown in (a). The lower (upper) boundary of the boxplots represents the 25th (75th) percentile, the mid-line in the box represents the median and the bottom and top of the vertical lines represent the 10th and 90th percentile, respectively.





**Figure 3.** Comparison of average (2012-2019) annual emissions (Tg a$^{-1}$) of VFEI and other inventories for different species in different regions shown in figure 2. The species compared are, from left to right, total carbon (C), carbon monoxide (CO), sulfur dioxide (SO$_2$), black carbon (BC), organic carbon (OC), and particulate matter < 2.5 μm (PM$_{2.5}$).






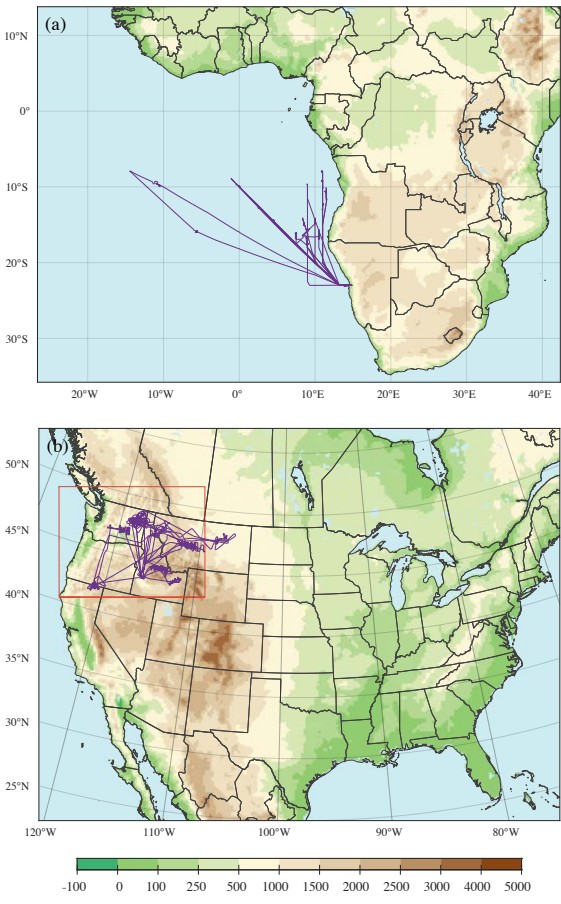

**Figure 4.** Domains of the simulations conducted in this study using WRF-Chem and VFEI (WRF-VFEI) showing topography (m.a.s.l.). Purple lines represent the flight tracks from ORACLES-2016 (Southern Africa) and FIREX-AQ (North America) used to evaluate the model results. The red polygon in (b) shows the inner domain used in this simulation.





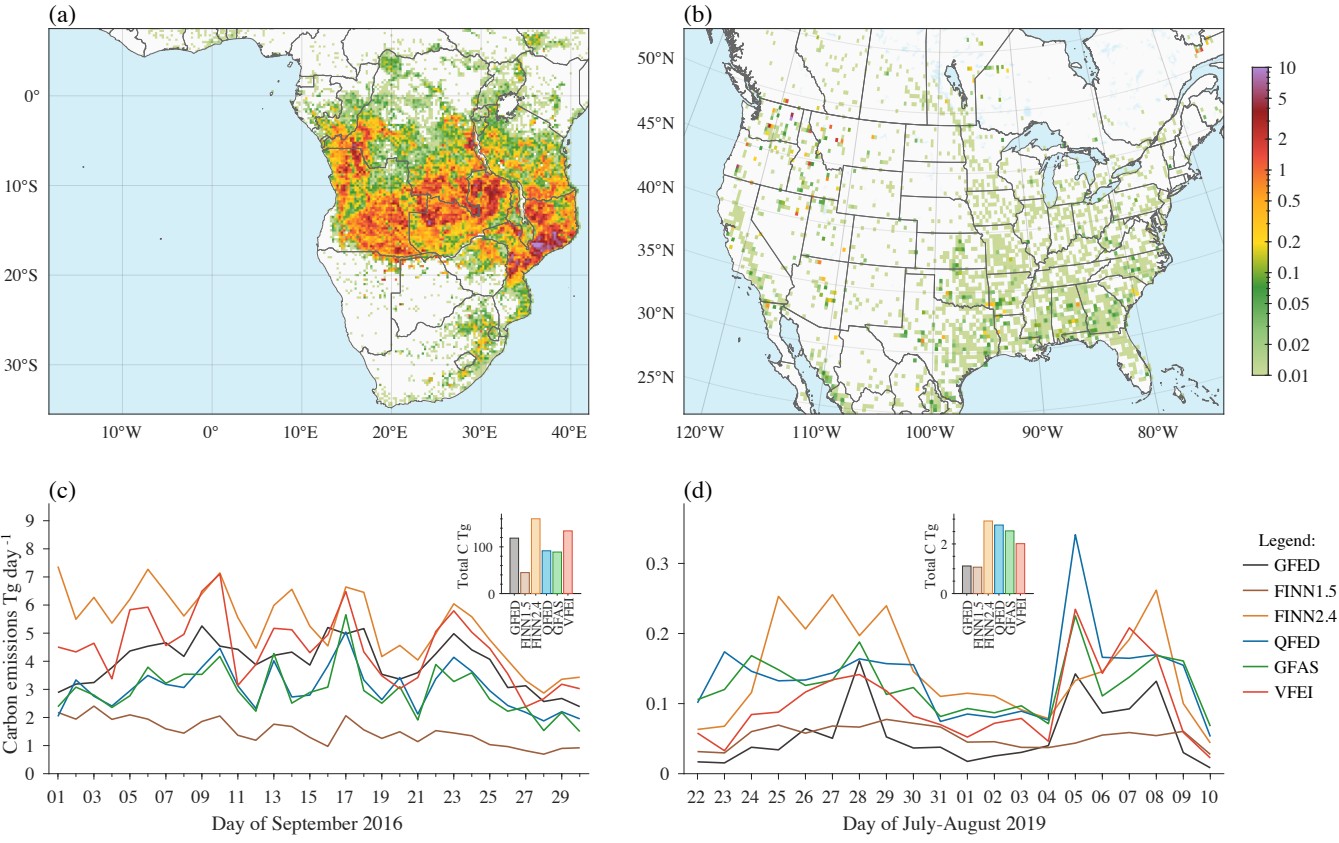

**Figure 5.** (Top) Total carbon emissions (Tg) for the period of (a) September 2016 and (b) between 20 July and 11 August 2019 from VFEI in each model simulation. (Bottom) Time series of daily carbon emissions (Tg day$^{-1}$) comparing VFEI to other inventories over the domains shown at the top. Small plots represent the total carbon emitted in each region during their period.





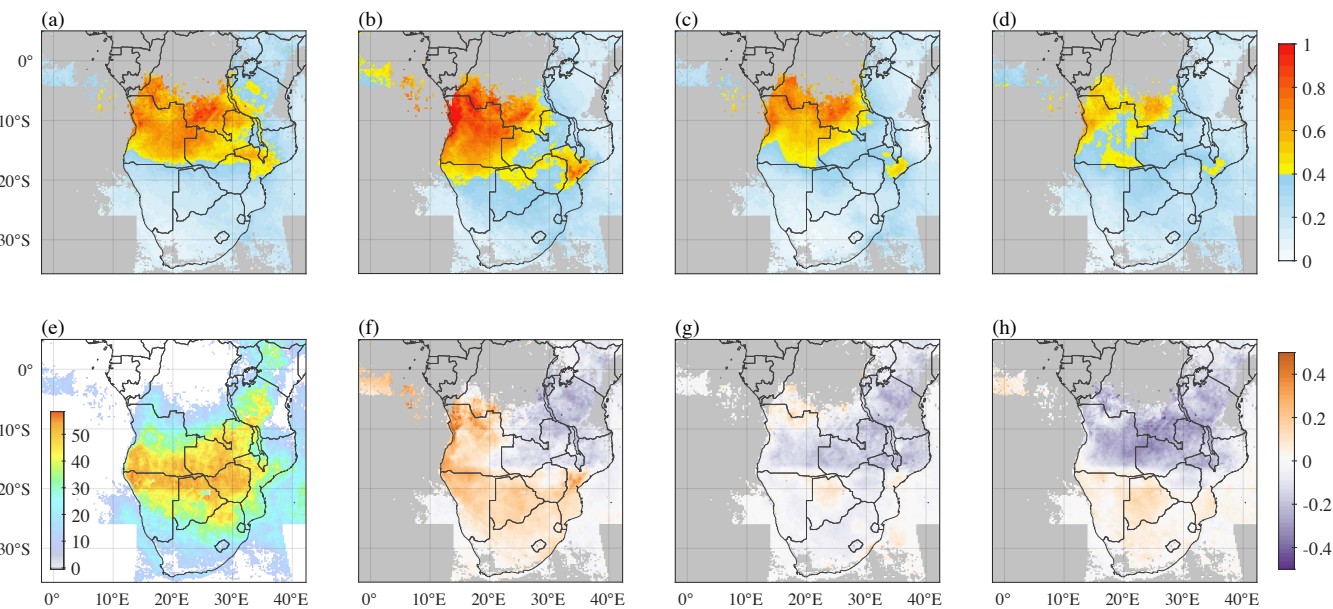

**Figure 6.** Monthly averaged aerosol optical depth at 550 nm from (a) MODIS MAIAC, (b) WRF-VFEI, (c) CAMS reanalysis and (d) MERRA-2 reanalysis during September 2016. (e) number of observations per grid point weighted in the final average for each dataset. Panels (f-h) show the mean monthly bias from each dataset with respect to MODIS MAIAC. Gray areas represent grid points with no observations during the period.



**Figure 7.** Comparison of aerosol optical depth (AOD) at 500 nm between AERONET sites and WRF-VFEI for September 2016. Panels (a-f) show the data as time series from six selected sites. AOD data at 550 nm from MODIS MAIAC C6.1 at the location of the sites is included as reference. Panel (g) shows the location of the AERONET sites used to evaluate the model simulations. The stations shown in panels (a-f) are highlighted in red. (h) Scatter plot displayed as heat map made using data from all the AERONET sites shown in (g). Statistics are reported in each panel, such as number of points (N), correlation (R), mean bias (MB), normalized mean bias (NMB) and root mean squared error (RMSE).

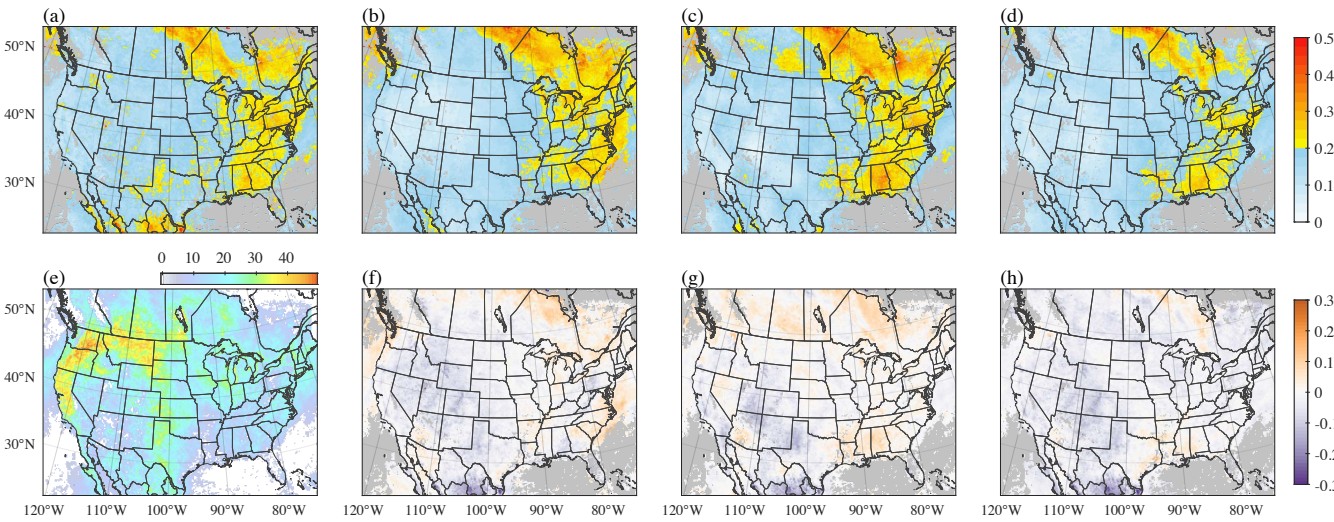

**Figure 8.** Aerosol optical depth at 550 nm from (a) MODIS MAIAC C6.1, (b) WRF-VFEI, (c) CAMS reanalysis and (d) MERRA-2 reanalysis averaged between 22 July and 10 August 2019. (e) number of observations per grid point weighted in the final average for each dataset. Panels (f-h) show the mean monthly bias from each dataset with respect to MODIS MAIAC C6.1. Gray areas represent grid points with no observations during the period.



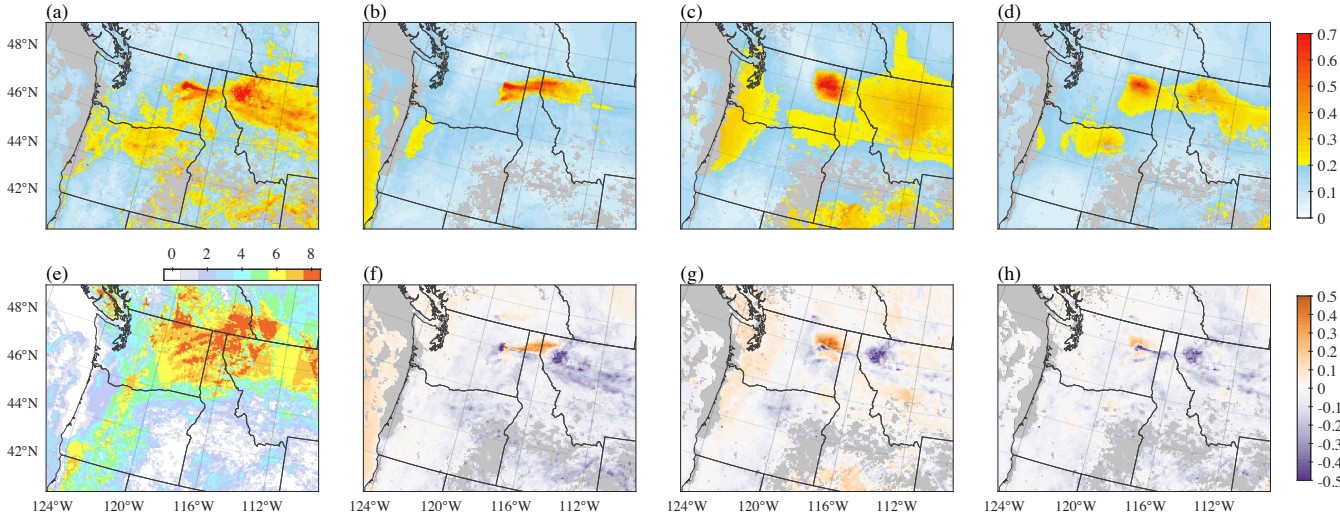

**Figure 9.** Same as in figure 8 but focusing on the inner domain of the simulation. The period averaged here is 6-9 August 2019, aiming on the Williams Flats fire event.







Figure 10. Same as figure 7 but comparing AERONET stations from North America between 22 July and 10 August 2019.



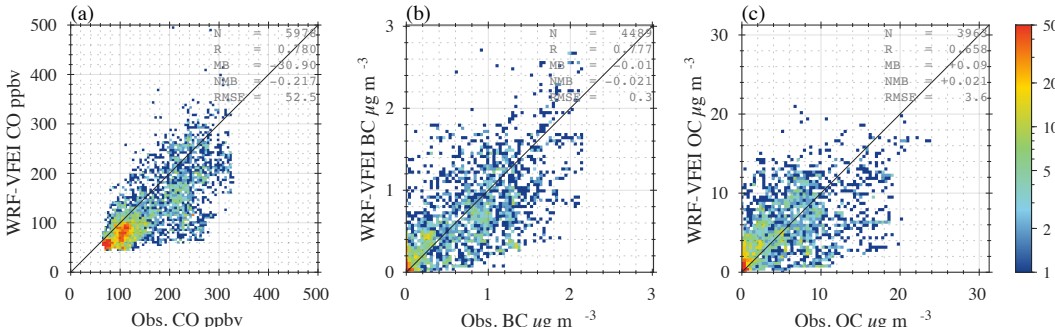

**Figure 11.** Scatter plots displayed as heat maps comparing the airborne data from the ORACLES-2016 campaign in Southern Africa and the WRF-VFEI simulations. (a) carbon monoxide in ppb, (b) black carbon in µg m⁻³ and (c) organic carbon in µg m⁻³. Statistics are reported in each panel, such as number of points (N), correlation (R), mean bias (MB), normalized mean bias (NMB) and root mean squared error (RMSE).





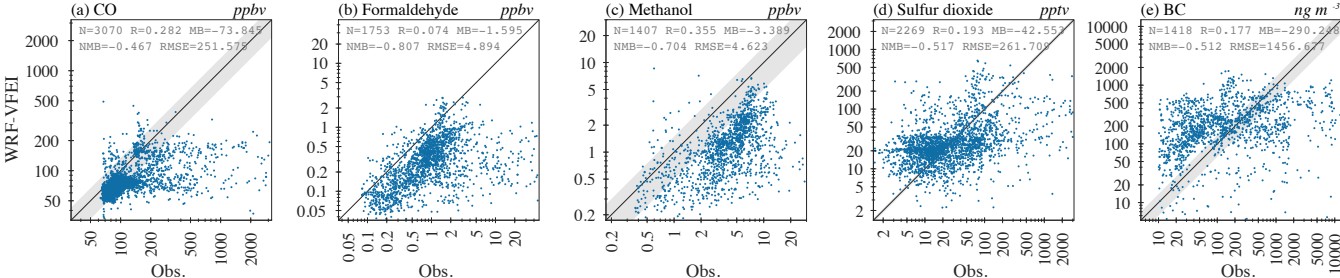

**Figure 12.** Scatter plots comparing the airborne data from the FIREX-AQ campaign in North America and the WRF-VFEI simulations between 24 July and 8 August 2019. (a) carbon monoxide in ppb, (b) formaldehyde in ppb, (c) methanol in ppb, (d) sulfur dioxide in ppt and (d) black carbon in ng m$^{-3}$. Statistics are reported in each panel, such as number of points (N), correlation (R), mean bias (MB), normalized mean bias (NMB) and root mean squared error (RMSE).

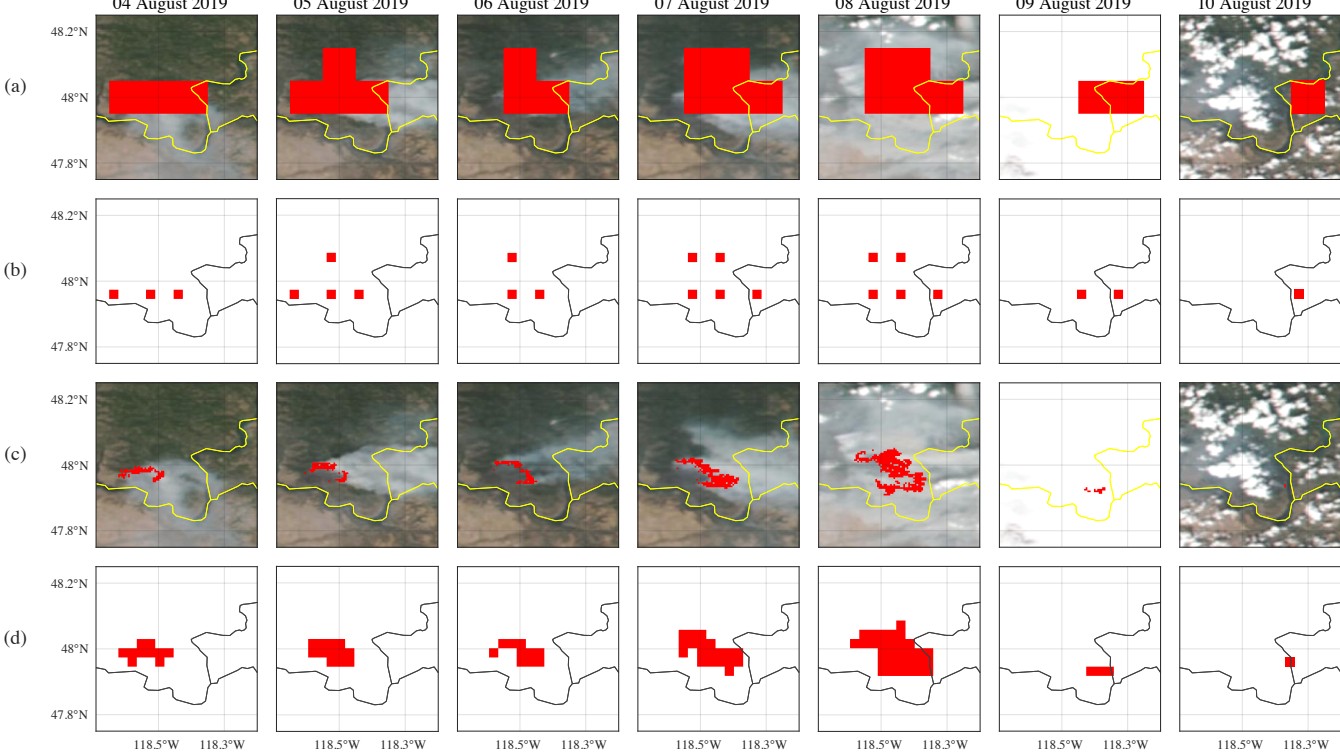

**Figure 13.** Example of mapping of a large fire event into a high-resolution (3 km) model grid. (a) VFEI emissions at 0.1° resolution. (b) VFEI emissions at 0.1° resolution mapped into high-resolution model grid. (c) VFEI at native ~500 m resolution. (d) VFEI at native ~500 m resolution mapped into the same high-resolution model grid. Each column represents a different day of early August 2019 shown at the top. Panels (a) and (c) include an overlay of VIIRS visible imagery from each day as reference. The VIIRS visible imagery were obtained from NASA Worldview Snapshots (https://wvs.earthdata.nasa.gov; last access: 21 April 2022).