# Peer review of "Introducing a VIIRS-based Fire Emission Inventory version 0 (VFEIv0)"

_Geoscientific Model Development, 2022_

## Author Comment (AC1)

**Introducing a VIIRS-based Fire Emission Inventory version 0 (VFEIv0)**

Gonzalo A. Ferrada et al.
gonzalo-ferrada@uiowa.edu

**Response to RC1**

We greatly appreciate your review and comments provided. These helped to improve this manuscript. All our responses are provided in line and in color blue.

In this manuscript, Ferrada and colleagues present a newly developed biomass burning emission inventory, the VIIRS-based Fire Emission Inventory version 0 (VFEIv0). There are multiple bottom-up (burned area based) and top-down (satellite based) biomass burning emission inventories out there, but VFEI does offer unique advantages, especially the high native horizontal resolution. Overall, this is a sound effort, and aligns well with the scope of the journal.

The source characteristics of wildfires is a long-standing issue and major source of uncertainty in representing wildfires in air quality models and chemistry-climate models. One key issue faced by the broader community is that, we have multiple regional and global biomass burning emission inventories available and the results are vastly different. Due to the lack of direct emission/flux observations, it is extremely challenging to evaluate the individual inventories. This study (and many others) uses AOD, but quite a few other processes are involved as well (dispersion and transport, aerosol chemistry and physics, etc) and all together they affect AOD. Frankly speaking I am not entirely sure what we can actually learn from all those inter-comparison studies. Put it differently, we are limited by observations more than ever. However, novel approaches reporting (direct) emission flux/rate estimates for wildfires are becoming available, for instance, Stockwell et al. (2022), Bela et al. (2022), Wiggins et al. (2021). Obviously these observations are limited to isolated regions and fire seasons, but these are very rare (direct) emission flux/rate estimates that actually offer direct scientific insight into this otherwise poorly contained issue. In light of this, it will greatly strengthen this manuscript if the authors can take advantage of these unique datasets. Stockwell et al. (2022) and Wiggins et al. (2021) are available for the FIREX-AQ period.

This manuscript is generally well written, although certain aspects of the data analysis can use some improvements. I particularly enjoy Section 5, in which the authors demonstrated how fire source may be misplaced if a coarse resolution inventory is used. I recommend this manuscript for publication after the following comments and concerns are addressed.

Thanks for your suggestions. We agree. The need for accurate fire emissions and modeling is highly overdue. However, as you mention, it is very difficult (and frustrating) not having enough and specific observational data to compare and evaluate VFEI and other inventories. Here, we compared against other four inventories, but as we mentioned in the manuscript, all these inventories are estimations. At the end of the day, these comparisons just show how VFEI lays with respect to others, but they do not provide an objective evaluation. To somewhat achieve a more quantitative and objective analysis we conducted model simulations. Due to computational resources available, we only focused in two regions of the world. Therefore, the performance of VFEI in other regions with high discrepancies among inventories (e.g., Siberia and Southeast Asia) is still undetermined. In addition, evaluating the emissions by a full Earth System model also has its own complications. To name a few: emissions may (or may not) be right, but then the other components of the model will modify the composition and transport of the smoke plume, which makes the evaluation against observations (airborne and satellite-based) harder, and even

unfair in some cases. Hopefully, more in-situ and satellite-based observations of smoke plumes and composition could be a reality in the future. They would serve to highly improve our estimations on different species.

Following your suggestion of comparing VFEI to the measurements by the papers you cited, we made a comparison using the Fuel2Fire dataset comparing 14 fires sampled during FIREX-AQ. Overall, VFEI estimates more emissions for these fires (around 12 % more carbon). This is now included at the end of section 4 (section 4.4.2).

Specific comments:

L32: Global warming refers only to the rising global mean temperature. It is my opinion that climate change is a broader and better term in this context which includes several other key changes along with the warming trend.

We changed the term to climate change.

L65: "…often provide data for a limited number of species…" Technically inventories can provide data for whatever compound/species as long as emission factor/ratio is available. Due to historical or legacy reasons some inventories report data for selected species only (e.g., more comprehensive measurements become available more recently; certain inventories were initially developed for certain models or chemical mechanisms). Even if the emission factor of a compound of interest is not reported in a certain inventory, usually one can easily apply some scaling factor or emission ratio. The uncertainty associated with such scaling is usually smaller than the uncertainty introduced from other processes. Therefore this is not really a limitation or issue, in my opinion. The high native horizontal resolution, however, is a major advantage of VFEI.

This is true. Inventories can potentially provide data for other species if they have emission factors for them. However, what we mean in this sentence is that in their final version of their product, some inventories provide data for a limited number of species. It is true that they potentially could, but that data is not provided in the final dataset that is available to the public (e.g., QFED).

To clarify this, we complemented this phrase: "BB emission inventories often provide data for a limited number of species in their publicly available datasets, …".

L66-67: "These can be a problem in model simulations since missing species can be highly volatile (e.g., hydrocarbons) and, thus, producing unrealistic results in smoke plume composition." I don't quite follow this. Please clarify.

To continue with the previous point where we stated that inventories often provide data for a limited number of species, we mentioned that this could lead to an inaccurate representation of smoke composition in the model. By missing some species, the chemical reactions that lead to the formation of secondary species, such as secondary organic aerosols or ozone, will be altered. To clarify this, we improved this sentence to: "The first can be a problem in model simulations since missing species can be highly volatile (e.g., hydrocarbons) and, thus, producing unrealistic results in smoke plume composition by, for example, misrepresenting the formation of secondary species (e.g., ozone, secondary aerosols)."

L75: It has been well documented that wildfire emissions do have diurnal variabilities, with major impact on emissions. E.g., Wiggins et al. (2020). The authors should acknowledge that not having sub-daily variability is one important limitation of this inventory. I do understand that it is challenging to develop a global biomass burning emission inventory with sub-daily temporal resolution.

The manuscript discussed the limitations of VFEI not having a diurnal cycle incorporated beginning in line 498. To this date, none of the other widely used inventories discussed here provide data at the sub-daily level in their final publicly available products.

In our model simulations, we used different approaches to try to minimize the impact of the lack of diurnal cycle by VFEI. In section 4.2, we specified that for the simulations over Southern Africa, we used a gaussian distribution with peak at 2 PM local solar time following the findings of Roberts et al. (2006). In North America, we used the GOES-16 and -17 fire product to constrain better the diurnal cycle of large fire events, including the Williams Flats fire that we discussed in the manuscript. However, the GOES satellites captured a limited number of fires, possibly due to their coarser sensor resolution. The diurnal cycle for the other fires (not detected with GOES) was constrained using the biome-dependent average diurnal cycle as reported by Li et al. (2019).

We acknowledged the limitations of these approaches throughout section 4 and 6. However, it is important to note that the use of VIIRS in the case of VFEI (and MODIS in the case of other inventories) is because these sensors provide data at high resolution and, thus, detect a higher number of small fires. It would be ideal to complement VFEI with geostationary satellite data to constrain the diurnal cycle of fires. However, since geostationary satellites have coarser sensor resolution, they often miss a large portion of small fires. Therefore, for the majority of fires, a prescribed diurnal cycle from literature would have to be provided.

We added the paper of Wiggins et al. (2020) as reference in this new version of the manuscript to enrich our discussion.

L95: Heil et al. (2010): this citation in the reference list is missing key details. Please refer to the journal guidelines on reference style.

We fixed the reference for this citation on the reference list. We also found some others that were not in the correct format.

L94-97: Please discuss any potential biases that may be introduced by using MODIS-based conversion factors and FRP derived from VIIRS. Previous studies (e.g., Li et al. 2018) show that VIIRS and MODIS FRPs are broadly comparable but they do show discrepancies in certain regions.

Li et al. (2018) compared the FRP from the fire products from VIIRS and MODIS as we referenced in the manuscript. Globally, FRP values between these two sensors are comparable, but some differences arise in some regions. Li et al. discussed that most of these differences could be explained by (i) the different architecture inherent to the sensors (MODIS vs. VIIRS) and (ii) the slightly different algorithm used in both to accommodate such differences. In general, they found that VIIRS can detect more small fires, especially on the edges of the swath due to less distortion, and that is one of the reasons we decided to make VFEI using VIIRS.

Indeed, we noticed differences when comparing the reported FRP values from both sensors. We applied the same methodology as for VFEI (data filtering, aggregation, etc.), but using the MODIS active fire product instead of VIIRS'. We found that (globally), FRP values from MODIS are 20% higher than the ones from VIIRS. Therefore, in VFEI we scale up the FRP values (and thus, emissions) by a factor of 1.2. This value is in line with the findings of Li et al. (2018). As per suggestion of referee #2, we explained this further in the new version of the manuscript with some speculations for these differences to occur. This is now reflected in section 2.4.

L133-135: Wouldn't this (simply taking an average between day and nighttime) create a systematic bias depending on overpass time? Please elaborate.

The average between day and night is done because very few fires are observed or detected during nighttime. For example, fire A is observed with a FRP of 15 MW during daytime and 5 MW during nighttime. Then, we assume that the daily average for that fire is 10 MW, and we calculate emissions based on that. Now, let's consider fire B with a FRP of 15 MW during daytime, but undetected during nighttime (i.e., FRP = 0 MW; probably because the fire is already extinguished). Then, fire B daily average FRP is 7.5 MW. It would be 'unfair' to calculate emissions on fire B based on the 15 MW and this could lead to the overestimation of its emissions. Therefore, we take the average between day- and nighttime observations. It is important to note, the VFEI aggregates the fires into a very fine grid of 0.005° resolution before taking this average in order to minimize any bias that could be produced by this approach. Having a diurnal cycle could eliminate the need to apply this assumption. As mentioned in one of our previous answers, we discuss this in the conclusions section.

L207: please briefly describe what improvements are introduced in FINNv2.4, which lead to a ~2x difference compared to FINN1.5 (Figure 3).

FINNv2.4 was recently developed by NCAR and they do not have a manuscript describing their modifications compared to previous versions yet. The information we put on our manuscript was obtained by limited personal communication with the PI of FINN. In general terms, they were testing different enhancement factors in some regions and some other changes in their methods to be more in line with the other inventories, since older versions of FINN were documented to underestimate emissions. They mentioned that they were aware of the differences of FINNv2.4 with respect to the other inventories and mentioned that FINN is still in development for a future version that could reduce these differences.

L232-236: please see my comment above regarding the recent biomass burning emission flux/rate measurements. Please consider comparing FVEI to Stockwell et al. (2022) and Wiggins et al. (2021).

We are now including the evaluation of VFEI using total carbon emissions as estimated by Fuel2Fire for 14 fires occurred in the Western USA during the FIREX-AQ campaign. This is included in the new version of the manuscript at the end of section 4.4.2.

Section 4.3: (1) Emission is just one piece of the puzzle. Multiple other factors have profound impact on AOD as well, for instance, the size distribution, optical properties, hygroscopicity of smoke aerosols. Please briefly discuss if WRF-Chem captures these properties of smoke particles. The comprehensive measurements from FIREX-AQ may be useful. (2) The authors only present model simulations using VFEI. Why not show a few more simulations with other emission inventories?

We acknowledge that the method to calculate AOD may alter the final result reported by the model. Beginning in line 314, we mentioned how WRF-Chem does this calculation in general. It relies on the Mie code to calculate optical properties. Upon initialization, WRF-Chem with MOSIAC use constant refractive indexes. At the end of the conclusions' section, we also discussed that new approaches to better estimate AOD and optical properties have been developed recently. However, these developments are still not included in WRF-Chem. Because of this, we also compared our model results with other species (such as CO) in order to provide a different insight of VFEI emissions to not rely only on AOD. It is important to note, that these simulations attempted to provide a more objective evaluation of VFEI. But - as mentioned in our response to the main comment- models have several other components that will alter the emissions and smoke, making an objective assessment more challenging.

We would have loved to make simulations using other inventories as well. Ideally, we would have conducted global simulations for at least one year, but the computational cost to do either of these largely exceeded our resources. It is important to note that running with full chemistry as we did is up to 7 times more expensive than just meteorology. In the conclusions section, we also hint that further evaluations for other regions are needed (e.g., Southeast Asia, Siberia, Amazon, etc.).

L369: Another specific reason that may at least in part explain the widespread model underestimation in AOD as shown in Figure 9 is the long range transport of Siberian smoke. See Johnson et al. (2021). If I remember correctly, the front of the Siberian smoke reached the Pacific Northwest on 3-4 August 2019. Does the boundary condition used in WRF-Chem includes Siberian smoke?

We prepared boundary conditions for our simulations using CAMS reanalysis from ECMWF. CAMS uses AOD data assimilation which would consider any smoke (or any other source of aerosols) into our domains. However, their vertical resolution is coarse in higher levels of the troposphere. If the correct altitudes of the aerosol layers are not well represented when feeding this data into our model simulations, it may end altering the transport and deposition rates of the smoke, among other effects. However, proving this is very difficult and out of the scope of our manuscript.

L399: The authors ought to be careful about the language. Many of the instruments are sensitive enough at such concentrations (e.g., hundreds ppb of CO), therefore the high resolution data may well reflect the real features rather than noise. I understand that the model won't be able to resolve such features, therefore moving averages like the authors did here makes sense. But the reason is not because of the measurements are filled with noise but simply because the model cannot resolve these features.

Thank you for the correction. We modified the text accordingly and now we state that we took 60s averages because the model simply will never be able to reproduce rapid changes in concentrations.

L404: "This may suggest that the background CO in the model is underestimated, rather than VFEI emissions themselves…" Well if the authors subtract the background and focus on only smoke, one will be able to tell if the bias is driven by background CO or fire emited cO.

We thought of subtracting the background CO when making the comparisons. The problem would be that doing so could be subjective and difficult to explain. Some of the questions we had when thinking on this were: do we consider background 10 ppb, 30 ppb, 50 ppb? Do we apply this only for Southern Africa and not North America? Do we subtract background for the other species that come from the boundary conditions as well or just CO?. In addition, subtracting CO from the model would also make difficult the comparison against the airborne data, since the data is reported as a total concentration of CO and it is not partitioned between background, fire, or anthropogenic sources.

Because of the arbitrary nature of this, we thought that it was better suggesting that the CO may be underestimated, and we included the figure S7 to support this idea. Figure S7 shows that the CO is systematically underestimated by the model in all the flights conducted in the domain across multiple regions.

It is important to note that our simulations included not only biomass burning emissions, but also anthropogenic and biogenic. We did this in order to fully represent the real state of the atmosphere in both domains.

Figure 11: Wildfires are challenging to represent in Eulerian models. Even with "perfect" emission inventories, a model may still underestimate the pollutant concentrations in fire plumes for the following reasons: numerical dilution (point sources); numerical diffusion; model places the fire source in the wrong grid(s); issues in plume rise. Therefore it is always challenging (sometimes unfair) to perform precise apple-to-apple comparison like in Figure 11. I would recommend that the authors show the ratios too, e.g., BC/CO, OC/CO, etc.

Our modeling results for North America were underwhelming when compared against the airborne data from FIREX-AQ, mainly because the model failed to reproduce the exact timing of fire activity and subsequent smoke development/transport. In our analysis, we conclude that most of the transects across the smoke plumes observed by the FIREX-AQ flights were simply not captured by the model, leading to noticeable biases. Because of this, we believe that showing ratios would not provide a clear conclusion either, since most of the points were simply outside smoke plumes in the model. This could lead to statistically meaningless results that would be difficult to support. However, we are conducting new studies with VFEI that could allow for a more specific results in this sense and we hope to present them in a new manuscript.

L413: Again, many of the the high resolution airborne measurements are sensitive enough for fire plumes and hence often reflect real features rather than noise. e.g. Palm et al. (2021), Wang et al. (2021)

We corrected this sentence as well.

L415: "WRF-VFEI produces accurate results when comparing CO concentrations of less than 150 ppb…" I wouldn't say accurate here. If the authors produce the same 2D distribution plots (same as Figure 11), most datapoints with <150 ppb CO would fall below the lower bound of the gray shading.

We modified this sentence to reflect these underestimations.

L417-418: "VFEI slightly underestimates the BB emissions over the North American Pacific Northwest during this period…" Well, without comparing to actual measurements of emissions, one cannot simply say this. Comparing to other inventories does not justify this statement since other inventories are not direct measurements either. Consider evaluating VFEI using Stockwell et al. (2022).

This sentence is included to explain that the biases of the model results could be explained by it and/or a combination of others mentioned later. We modified this sentence to be more accurate and that also is in line with the new results we are including in this new version of the manuscript, which includes a comparison between the Fuel2Fire and VFEI carbon emissions.

L423: "…using the nearest grid point to the trajectories of the flights, even a slight misrepresentation of the smoke can introduce large biases in the results" This is indeed a simple and widely used approach. I'd argue that using nearest 2D interpolation may reduce this impact. But I also agree that it is challenging for a model with coarse resolution to resolve fine features.

Yes. Models report the instantaneous concentrations, but they represent the 3-D average of each grid box. This is one of the instances where the model resolution may have an impact on results and biases observed. Models also have monotonic advection which 'smooths' abrupt gradients. These subjects are inherent to all models.

Figure 12: i'm curious why the authors do not show OM or OC, which accounts for the majority of smoke aerosol mass. Figure 11 does show OC. Also following my previous comment: it may be challenging for the model to capture the exact mass concentrations, but the ratios tell us more about the modeling system. For instance, Figure 12 tells me that CO is severely underestimated but less so for BC, implying that BC/CO ratio is perhaps overestimated, which indicates an issue in emission factors, fuel categories, treatment of burning condition, etc. Also the impact of anthropogenic pollution will be reduced by subtracting the background. This is necessary for CO since the CO levels in the background air are usually quite substantial.

We configured our model simulations to provide a limited number of species and variables to optimize storage space. This is why we did not make a more extensive evaluation using the dozens of observations for different species provided by the FIREX-AQ campaign.

As we mentioned in one of our earlier responses, removing the 'background' CO could be tricky and challenging to explain and convince reviewers. We had this in mind when conducting our simulations. Therefore, we used not only VFEI emissions to account for biomass burning, but also considered biogenic and anthropogenic emissions. The last being the National Emission Inventory (NEI) prepared by the US Environmental Protection Agency in 2017.

Results for North America were underwhelming compared to Southern Africa's. Specially because the modeling system did not reproduce the correct timing on the release of VFEI emissions and the subsequent smoke plume development and transport. Because of this, we did not do further analysis with this data. As we reflect in the manuscript, we believe this can be largely improved by including an individualized treatment to the diurnal cycle of fires.

Section 5: this is brilliant! Nicely designed and demonstrates the advantage of VFEI.

Thank you!

References

Stockwell et al. 2022. https://pubs-acs-org.colorado.idm.oclc.org/doi/10.1021/acs.est.1c07121

Bela et al. 2022. https://agupubs.onlinelibrary.wiley.com/doi/10.1029/2021GL095831

Wiggins et al. 2021. https://agupubs.onlinelibrary.wiley.com/doi/full/10.1029/2021JD035692

Wiggins et al. 2020. https://agupubs.onlinelibrary.wiley.com/doi/10.1029/2020GL090707

Li et al. 2018. https://agupubs.onlinelibrary.wiley.com/doi/full/10.1029/2017jd027823

Johnson et al. 2021. https://www.sciencedirect.com/science/article/pii/S1352231021000595

Palm et al. 2021. https://agupubs.onlinelibrary.wiley.com/doi/full/10.1029/2021GL095443

Wang et al. 2021. https://agupubs.onlinelibrary.wiley.com/doi/full/10.1029/2021JD035203

---

## Author Comment (AC2)

**Introducing a VIIRS-based Fire Emission Inventory version 0 (VFEIv0)**

Gonzalo A. Ferrada et al.
gonzalo-ferrada@uiowa.edu

**Response to RC2**

We thank the reviewer for their time invested in reviewing this manuscript. Your comments were very helpful, and we incorporated most of the suggestions in the revised version of the manuscript.

We responded to the specific comments in line and in blue.

Biomass burning emissions are one of the large sources of uncertainty in modeling atmospheric composition. This paper describes a new fire emissions inventory that is based on fire radiative power measured by a relatively new satellite instrument (VIIRS), with high spatial resolution. There is a very wide range in emissions estimated by available biomass burning inventories, and this new inventory generally falls in that range. This work is valuable for providing another global inventory at higher spatial resolution than some others. It also contributes to quantifying the uncertainty in fire emissions by using slightly different input data sets and procedures than other inventories.

The paper is clearly written, and the creation of the inventory is described well. The resulting emissions are available on a public website in netcdf files, so readily usable by the community.

The emissions are compared to other inventories in clear plots. Simulations with WRF-Chem using the new emissions, for a couple of regions (N.America, S. Africa) for short time periods, are evaluated with observations and show reasonable performance.

Listed below are a few technical corrections and suggestions for clarifying the text. There are a number of preposition errors (in/on/at) that I have not bothered to list, so additional proof-reading is recommmended.

l. 29: 'are used' -> 'are used for'

Done.

Lines 42 & 202 & Table 4: FINNv1.5 does not use MODIS Burned Area products. Fire locations are determined from MODIS Thermal Anomalies, and the size of the fire is determined by other means. See Wiedinmyer et al 2011 for details on how the area of each fire is determined.

We modified the description of FINN following Wiedinmyer et al. (2011).

l.119: mid-202 -- which year?

Mid-2021. We corrected this.

l.129: hotpots -> hotspots

Corrected.

l.168-171: This is not explained well; I do not understand how the factor 1.2 is found. The statement "using the same methodology described here" is unclear.

The factor 1.2 was found by using the MODIS active fire product (instead of VIIRS') and using it to generate emissions with it. This is, we applied the same methodology as described for VFEI (data filtering, aggregation, etc.). Globally, we found that the FRPs from MODIS were 20% higher than the ones from VIIRS. In the text we speculate that this may be due to the fact of having a double number of observations from MODIS (Terra and Aqua) when compared to the VIIRS onboard of Suomi-NPP. Hence, VIIRS may be missing the detection of some fires occurring at other times, among other possible causes.

In our first version of the manuscript, we did not explain this further, because these are basically the same findings of Li et al. (2018). The revised version includes a better explanation on how we found this factor and the limitations of this method.

l.199: 'other four' -> 'four other'

Done.

l.200: 'rely in' -> 'rely on'

Fixed.

Fig. S1: reference to Figure X.

Corrected. It now references to Figure 1.

l.418, 420: Figure 6d -> 5d; 4c -> 3c

Fixed. Thanks!

l.436: FINN also provides emissions for each 1-km fire.

Added to the list.

l.439: 'If VFEI would have' -> 'If VFEI had'

Corrected.

Section 5 and Fig. 13: I do not follow the reasoning that the fire emissions at 0.1 degree are necessarily represented in a 3-km grid as shown in Fig.13b. Some models (modelers) would use the 0.1-deg emissions in all the 3-km grid boxes that fall within that box (of course conserving mass). I do not dispute the value of having the finer resolution representation of fire emissions that VFEI provides, but I would guess that the step from panel (a) to panel (b) is not universally, or even generally, applied. The discussion should be qualified, or perhaps just leave out row (b) of Fig. 13.

We modified this section to reflect your reasoning. What you state is true: there are different ways to address this issue but the figure (as its caption indicates) and the purpose of this section is just to give an example. Distributing the emissions within all the grid boxes that fall within the coarser resolution inventory would not be much of a problem for the case shown in the example (the Williams Flats fire was a single large fire event that covered several grid boxes). However, applying this as a standard method for all fires could create unrealistic results for the case of small fires of -for instance- 1 hectare with no other fires around (e.g., agricultural fires in Europe).